# Glucagon-Like Peptide-1 (GLP-1) in the Integration of Neural and Endocrine Responses to Stress

**DOI:** 10.3390/nu12113304

**Published:** 2020-10-28

**Authors:** Yolanda Diz-Chaves, Salvador Herrera-Pérez, Lucas C. González-Matías, José Antonio Lamas, Federico Mallo

**Affiliations:** 1CINBIO, Universidade de Vigo, Grupo FB3A, Laboratorio de Endocrinología, 36310 Vigo, Spain; lucascgm@uvigo.es; 2CINBIO, Universidade de Vigo, Grupo FB3B, Laboratorio de Neurociencia, 36310 Vigo, Spain; ssalva4@me.com (S.H.-P.); antoniolamas@uvigo.es (J.A.L.)

**Keywords:** glucagon-like peptide-1, hypothalamic-pituitary-adrenal (HPA) axis, sympathetic nervous system (SNS), ion channels, food intake

## Abstract

Glucagon like-peptide 1 (GLP-1) within the brain is produced by a population of preproglucagon neurons located in the caudal nucleus of the solitary tract. These neurons project to the hypothalamus and another forebrain, hindbrain, and mesolimbic brain areas control the autonomic function, feeding, and the motivation to feed or regulate the stress response and the hypothalamic-pituitary-adrenal axis. GLP-1 receptor (GLP-1R) controls both food intake and feeding behavior (hunger-driven feeding, the hedonic value of food, and food motivation). The activation of GLP-1 receptors involves second messenger pathways and ionic events in the autonomic nervous system, which are very relevant to explain the essential central actions of GLP-1 as neuromodulator coordinating food intake in response to a physiological and stress-related stimulus to maintain homeostasis. Alterations in GLP-1 signaling associated with obesity or chronic stress induce the dysregulation of eating behavior. This review summarized the experimental shreds of evidence from studies using GLP-1R agonists to describe the neural and endocrine integration of stress responses and feeding behavior.

## 1. Introduction

Population-based and clinical studies data point out a significant and positive association of chronic stress states and big uncontrollable stressful events with body mass index (BMI) and weight gain [1]. In both people and animals, regardless of total caloric intake does not increase with stress; a shift toward choosing more pleasurable or palatable calories occurs [2]. Foods rich in sugars and fat are potent rewards, interacting to potentiate reward and engagement of neural circuits involved in habit formation and reward value [3] and trigger learned associations between the stimulus and the reward (conditioning) [4]. In this respect, stress becomes a critical risk factor affecting both the development of addictive disorders and relapse to addictive behaviors [1].

It has become increasingly clear the existence of brain networks’ integrated activity in controlling feeding behavior. The homeostatic control of feeding is regulated by hormones that control hunger, satiety, and adiposity levels and act on hypothalamic and brainstem circuits to maintain appropriate energy balance [5,6]. The brain reward systems also play an essential role in feeding behavior, and the mesocortical dopamine system is crucial in these reward-related processes [2]. In this regard, the ventral tegmental area (VTA), and the adjacent substancia nigra pars compacta, receive energy-balance information via orexin-containing projections from the lateral hypothalamus (LH) [7,8]. From VTA/SNc arise dopamine signals that innervate the nucleus accumbens (NAc) and dorsal striatum, areas that regulate the motivational and incentive properties of food [9]. Nevertheless, dopamine projections into the dorsal striatum, cortical, and limbic regions encode information related to food’s reward value [10]. Moreover, the lateral hypothalamus is critically involved in food-seeking behavior, integrating essential metabolic signals with upstream and downstream behavioral effector circuits [4].

Satiation signals such as gastrointestinal peptides released by food ingestion, emerge as promising therapeutically tools in controlling feeding and obesity. For decades it has been known that glucagon-like peptide-1 (GLP-1) reduces food intake, acting as a short-term prandial signal [11]. However, also GLP-1 produced in the brain is involved in a satiation/satiety circuit controlling food intake and body weight [12,13]. GLP-1 is a member of the glucagon peptide superfamily, continuously secreted by the enteroendocrine L-cells at low basal levels in the fasting or interprandial state [14,15]. Meal intake promotes a rapid increase in L-cell secretion [11] depending on the meal’s magnitude and strongly correlated to gastric emptying [16]. GLP-1 is released by nutritional components stimulation, such as simple carbohydrates (glucose, fructose, and galactose), amino acids, protein, and fatty acids [17]. GLP-1 is synthesized from the proglucagon (Gcg) gene, after cell-specific post-translational processing [18] in L-cells, pancreatic islet α-cells, and brain, by two members of the convertase subtilisin/kexin family [19,20].

The primary source of endogenous GLP-1 within the brain is a population of preproglucagon-neurons (PPG) in the caudal portion of the nucleus of the solitary tract (NTS) [21,22]. Ablation of these neurons in mice reduces active GLP-1 by 60% in the brainstem and almost 80% in the hypothalamus and spinal cord [13,21,23]. Cell bodies of PPG neurons also have been described in the adjacent medullary reticular formation, both in rodents, non-human primates, and humans [21,22,23,24,25,26], as well as in a small population of glutamatergic olfactory bulb interneurons that express PPG that can modulate the firing pattern of the mitral cells in rodents [13,22,23,27,28]. Axons of these neurons are widespread throughout the NTS, the dorsal vagal nucleus, and the reticular nucleus (except for the parvicellular region) [22]. Their axons extend to the area postrema (AP) and the dorsal vagal nucleus [22]. Also, rostral to the medulla, in the Barrington’s nucleus and the locus coeruleus, there are many axons from PPG-neurons [22]

Immunohistochemical, hybridization in situ studies in rats and mice, and likewise non-human primates and the use also of selective fluorescent protein expression by a cre-dependent adeno-associated virus in mice have shown that PPG neurons project widely to regions that express the GLP-1 receptor (GLP-1R) in the central and autonomic nervous systems [21,22,29,30,31]. The hypothalamus receives a massive input from PPG-neurons, the majority to either the paraventricular (PVN) or the dorsomedial (DMH) hypothalamic nuclei [21,30,32,33]. However, GLP-1R nerve fibers are observed throughout the hypothalamus in rodents with a notable difference in the arcuate nucleus (ARC) versus primates. This area receives the densest innervation of GLP-1 immunoreactivity input in the primate brain [32,33].

Furthermore, these PPG-neurons project to other brainstem areas, in which different autonomic neurons have been identified, including catecholamine and serotonin neurons [29]. Also, the limbic forebrain regions [22,29,34], and areas of the mesolimbic reward system related to control of feeding and motivation to feed, such as the VTA, the NAc, the parabrachial nucleus (PBN), or the suprammammillary nucleus [23,31,35,36,37], or structures implicated in the regulation of the stress response and the hypothalamic-pituitary-adrenal (HPA) axis are innervated by NTS PPG-neurons [38]. In this regard, substantial numbers of PPG axons are present in mesolimbic areas of the forebrain, with the highest density of PPG axons observed in the bed nucleus of the stria terminalis (BNST) [33], at low to moderate density of these axons are identified in the dorsal and ventral parts of the lateral septum (LS) proximal to the lateral ventricles, the medial septum (MS) and occasionally the septohippocampal nucleus (SHi) [33]. The central nucleus (CeA), medial nucleus (MeA), and extended amygdala (EA), however, revealed a low to moderate density of PPG axon innervation [33]. Moreover, sympathetic preganglionic neurons are also innervated by PPG-neurons [39].

GLP-1 has a broader range of pleiotropic physiological effects, including the inhibition of the glucagon secretion [40] and gastrointestinal secretion and motility. However, it also displays other heart and blood vessels’ actions, lung, ovaries, gut, liver, immune cells, kidney, white adipose tissue, skeletal muscle, and brain [41,42,43,44,45,46]. GLP-1 regulates brain areas that modulate food reward with particular physiological relevance in controlling feeding behavior [47]. Moreover, GLP-1 plays a major role in overall physiological processes in response to stress [48,49], and this peptide is critical in maintaining energy homeostasis controlling satiety and hedonic aspects of food intake concerning stress [13,50,51,52]. GLP-1R agonists (GLP-1RAs), such as exenatide, liraglutide, albiglutide, dulaglutide, and recently semaglutide, approved for the treatment of type 2 diabesity and obesity, have become a promising pharmacological tool for reducing food intake and body weight [53,54] and new as yet unrecognized therapeutic indications could be developed.

In this review, we summarized what is currently known about the involvement of GLP-1 in the HPA axis regulation, food intake control, stress responses, modulation of sympathetic activity, and especially the ionic events in the hypothalamic neurons that might explain some of the most relevant biological actions of GLP-1 (Figure 1).

## 2. GLP-1 Receptor and Signaling Pathways

GLP-1R has sequence homology with the receptors for secretin, calcitonin, and parathyroid hormone, forming class B, a family of G-coupled receptors [55,56]. GLP-1R mRNA is expressed in pancreatic islets; predominantly in β-cells (~80% of the islet population), non β-cells (~20%) also express GLP1R (α-and (~12%) and δ-cells), lung, stomach, heart, ovary, kidney, 3T3-L1 preadipocytes and in mouse and human mature adipocytes, also in human epicardial fat and in visceral and subcutaneous adipose tissue [46,57,58,59,60,61,62,63,64,65,66]. Despite this, there may be differences in GLP-1 effects among species since dogs express its receptor in muscle and adipose tissue and mice also in the liver [67,68]. In the rodent brain, regardless of the appearance of some differences in receptor expression between rats and mice, depending on the technique of analysis employed, several studies using chemical biology, recombinant genetics, and super-resolution compatible labeling probes methods had described high content of GLP-1R in the hypothalamus (preoptic area, PVN, supraoptic, arcuate, DMH, lateral and ventromedial nuclei), also in the circumventricular organs such as the AP, and the choroid plexus (CP). In the arcuate, AP, and CP, super-resolution snapshots show that GLP-1R appears organized as nanodomains at the membrane of GLP-1R positive neuron cell bodies, as well as dendrites, in mice [66]. Also, GLP-1R expression appears in cells in the posterior thalamus, medulla oblongata, and pituitary, LS, amygdala (mainly in mice), NAc (the core, the shell), BNST, VTA, dorsal nucleus of the vagus; lateral reticular nucleus, and spinal cord, olfactory bulb, or supramammillary nucleus (SuM), among others [23,66,69,70,71,72,73,74].

Similarly, in non-human primates, the GLP-1R mRNA and protein expression present similar distribution in the CNS to rodents [75]. Also, in the human brain, mRNA was found in the cerebral cortex (especially in the occipital and frontal cortex), hypothalamus (mainly the ventromedial and arcuate nuclei), hippocampus, thalamus, caudate-putamen, and globus pallidum [40]. The GLP-1 receptor is expressed in neurons and glial cells [41,76,77,78]. In this regard, immunohistochemical studies demonstrated that GLP-1Rs are expressed in the CA region’s pyramidal cell layer and the granule cell layer of the dentate gyrus in the hippocampus pyramidal neurons in the neocortex and Purkinje cell in the cerebellum. Preferably in the dendrites of larger neurons, indicating its expression near synapses [76]. Moreover, astrocytes [42,79] and microglia [77,80] express GLP-1R.

Furthermore, the receptor of GLP-1 is expressed by the vagal afferent neurons that innervate the abdominal organs (gastrointestinal tract), including the hepatoportal region [45,46], and also within the nodose ganglion (NG) [46]. It should consider that the use of antibodies for visualizing the GLP-1R has limits of detection and possesses variable specificity and tissue penetration. Moreover, fixation techniques in different cells and tissues can affect the epitopes, that also can be hidden, leading to the ambiguous identification of GLP-1R-expressing cells in humans and animal species [81].

The activation of GLP-1R both centrally and peripherally induces an increase in intracellular calcium (Ca^2+^)i [27,82], and evoke an increase in cAMP production [83]. GLP-1 effects are generally slow (minutes range), supporting the hypothesis of an indirect activation through second messenger pathways [11,82,84,85]. Since forskolin, an activator of adenylate cyclase, also increases intracellular calcium and Rp-8-Br-cAMP, a protein kinase A (PKA) inhibitor, prevents the effect of GLP-1 in β-cells [86] and channels expressed in cell lines (heterologous systems) [87], it has been suggested that GLP-1 induces the activation of the cAMP pathway [82,88,89]. Therefore, PKA’s subsequent activation appears as the canonical signaling pathway accepted for inducing insulin secretion [90]. However, there exist other cAMP mediators, namely Epac [91,92] and other pathways like the inositol 1,4,5-trisphosphate (IP3), that also could be involved in the GLP-1 action [89,92,93,94]. However, each of these putative pathways’ contribution remains poorly characterized, which is essential for understanding GLP-1 effects. For example, and as see below, the understanding of the pro-GABAergic action of the activation of GLP-1R is necessary to explain how GLP-1 exerts different functions in the peripheral organs and the central nervous system.

### 2.1. Molecular Effects of GLP1 in the Hypothalamic Area

Most of the central actions of GLP-1 analogs in the nervous system are linked to changes in the activation of different groups of neurons, some of which are housed in the hypothalamus. Just like in pancreatic cells [87,95], the activation of GLP-1 also modulates the electrical properties in hypothalamic neurons. Many hypothalamic neurons are specialized in the secretion of neuropeptides involved in regulating the neuroendocrine axis, modulating the pituitary’s hormonal secretion. The studies in these neurons have extensively focused on intracellular second messenger signaling cascades, as discussed above. However, in addition to the activation of these second messenger pathways, the presence of GLP-1 causes changes in resting membrane potential (RMP) in these neurons. Specifically, the presence of GLP1 or analogs exerts significant depolarization making them more excitable [96] in hypothalamic neurons, including an increment in the firing rate in gonadotropin-releasing neurons (GnRH) at the hypothalamus [97]. Partly because of the reduction of the afterhyperpolarization current [96] (Table 1).

Since the RMP depolarization can be produced both by activation of the inward current (Na^+^ or Ca^2+^) and by blocking of outward current (K^+^), in the hypothalamus, several ionic channels may be involved in the depolarizing effect observed in the presence of GLP-1 and analogs. Table 1 summarize these effects.

#### 2.1.1. Voltage-Gated Na^+^ Channels

The substitution of sodium for choline in the culture media avoids the depolarizing effect of Exendin-4 (Ex4) in hypothalamic neurons, indicating that GLP-1 induced depolarization may be due, at least in part, to an effect on voltage-gated sodium channels (N_av_) [96]. However, the depolarizing effect in the hypocretin (Orexin-A)-releasing neurons can be suppressed by glutamate and GABA receptor blockers [96], suggesting the participation of G-proteins rather than a direct effect on N_av_ channels [98] by GLP-1R agonists.

#### 2.1.2. Voltage-Gated Ca^2+^ Channels and Nonselective Cationic Current

GLP-1 increases intracellular calcium in NG neurons [82], and the influx of Ca^2+^ has been ascribed to the activation of L-Type voltage-gated Ca^2+^ channels [99,100,101]. Additionally, in hypothalamic slices, the application of Ex-4 results in an inward barium current (*I*_Ba_) with a reversal potential around −30 mV, and thus supports the hypothesis of GLP-1 modulating a nonselective cationic current in the hypothalamus [96]. 

Even though the evidence comes from non-neuronal cells, the contribution of intracellular Ca^2+^ storages is unclear [102], but GLP-1 contributes to increasing IP3 levels [92,93,94], and this could contribute to the rise in cytosolic calcium that occurs in the presence of GLP-1 [103].

#### 2.1.3. K^+^ Channels

When the levels of glucose in the blood are low, the ATP-dependent potassium channels (K_ATP_) of pancreatic β-cells remain open and keep the membrane potential at a hyperpolarized level (around −70 mV) and hence in a non-secreting state (rest) [104]. Several types of voltage-dependent potassium channels are affected by GLP-1. Although GLP-1 inhibits an A-type potassium current (*I*_A_) in peripheral cells [105], Goycolea et al. failed to block *I*_A_ in the presence of GLP-1 and other analogs in hypothalamic slices [96]. 

TREK channels are members of the two-pore domain potassium channels (K2P) superfamily and widely expressed in several tissues such as the hypothalamus, among others. This family of K2P channels comprises three members, TREK-1, TREK-2, and TRAAK [106]. These channels are blocked by cAMP [107,108,109]. Knockout mice for TREK-1 (TREK-1^−^/^−^) have a phenotype resistant to depression and stress, and the activation of the HPA axis in response to induced stress results in a lower corticosterone level, indicating a close relationship between TREK and stress in this mice model [110]. Since GLP-1R activation results in increased cAMP levels [104,111] and glucocorticoids also increase the cAMP level in mice and humans in β-cells [112], it could speculate that the increase in cAMP might modulate the hypothalamic TREK channels and make hypothalamic cells more excitable.

On the other hand, anxiolytic and antidepressant compounds (such as fluoxetine, ostruthin, and spadin) are useful inhibitors of TREK channels [113,114,115,116]; similarly, it has been suggested that GLP-1 and its analogs (such as Ex-4 and Liraglutide) had anxiolytic and antidepressant effects [117,118,119]. Incretins are related to the activation of the HPA axis [120,121] too. Altogether, it is tempting to hypothesize that the positive (anxiolytic) effect of GLP-1 on stress responses may be related to the inhibition of TREK-1, and thus the interaction between TREK channels and the GLP-1 receptor must be investigated in deep. However, this fact could be controversial since some studies have shown how the activation of GLP-1R can have anxiogenic effects [37,120,122]. Therefore, this should be studied more thoroughly.

## 3. GLP-1 and the Stress Responses

Living organisms can survive because they maintain dynamic homeostasis, continuously challenged by internal or external adverse effects, termed stressors [123]. Different aversive physiological stimuli such as hypoglycemia, hypotension, hypoxia, hypovolemia, hypothermia, infection, and also psychological stressors, elicit a response that is characterized by the activation of the autonomic sympathetic nervous system (“fight or flight” response), that facilitates the secretion of noradrenaline and adrenaline by the adrenal medulla [124]. This response elicits rapid modifications in physiological states through neural innervation of end organs (increasing heart rate and blood glucose among others), and it is counter regulated and compensated by the parasympathetic (“rest and digest”) nervous system [38,125]. Moreover, the stressors also activate the HPA axis and stimulate de synthesis and secretion of glucocorticoids from the adrenal cortex [124,125,126]. In the context, GLP-1 emerges as a critical neuromodulator that mediates the response to stressors [120].

Although the stress response is critical for survival in the short term, dysfunctional stress responses are linked to several somatic and psychiatric diseases, affective disorders, and neurodegenerative diseases, emphasizing the importance of precise neuronal control of effector pathways [124,125,127]. 

### 3.1. GLP-1 Activates Hypothalamic-Pituitary-Adrenal Axis 

Activation of the hypothalamus-pituitary-adrenocortical axis represents a primary hormonal response to homeostatic challenge [128]. The HPA axis response’s direct-drive is mainly neuronal, initiated via excitatory neurotransmission at the PVN corticotropin-releasing factor (CRF)-neurons [128,129] by multiple circuits in forebrain limbic regions, hypothalamus, and brainstem [38]. The NTS is a crucial region in the processing of autonomic and HPA axis stress response both in the acute and chronic domains [38,129]. Catecholaminergic neuron fibers from the A2/C2 region in the NTS innervate CRF-neurons in the medial parvocellular part of the PVN [130]. These cells represent only a subset of stress-activated PVN-projecting neurons since non-catecholaminergic GLP-1 producing PPG-neurons also project to the PVN [21,131]. GLP-1-IR nerve endings or fibers and mRNA expression of GLP-1 receptor also exist in the PVN [132,133]. Similarly, mRNA for arginine vasopressin (AVP) and oxytocin is colocalizing in the supraoptic and PVN [134].

GLP-1 is involved in the HPA axis activation. The intracerebroventricular injection of GLP-1 induces c-fos expression in the supraoptic nucleus (SON) in the medial parvicellular subregion, and the magnocellular neurons of the PVN, predominantly in CRF-positive neurons [135,136]. Moreover, the administration of GLP-1 into the third ventricle, or the GLP-1 (7-36)-amide (iv), activate the HPA axis by increasing adrenocorticotropic hormone (ACTH) [120], AVP [135], and corticosterone (CORT) plasma levels [120,135], in a time-dependent manner, in conscious freely moving and anesthetized rats [121]. This effect was observed in humans, too, with an increase in cortisol secretion [121]. Likewise, central i.c.v. or peripheral administration of the GLP-1R agonist, Ex-4, actively elevate circulating ACTH and corticosterone levels in rats, potently activating the HPA axis [137,138]. Additionally, conditional deletion of GLP-1 receptor signaling in the PVN reduces HPA axis response to acute and chronic stress [139]. Accordingly, the reduction of stress response after GLP-1 antagonism or PVN deletion [129]. In this regard, i.c.v. administration of a GLP-1 antagonist, before placement of the rat in an isolated open arm of the elevated plus-maze (EPM), blocks the effect of the EPM to increase plasma ACTH and CORT and decreases the anxiety-like behaviors in response to stress in this task [120].

PVN and CRF could emerge as the principal mediators of GLP-1 effects on the HPA axis. The central blockade of CRF receptor type 1 with the nonselective CRF receptor antagonist, astressin, attenuates GLP-1-induced elevations of ACTH and corticosterone in plasma [120]. On this point, the mechanism of action of Ex-4 in the stimulation of the HPA axis activity depends on the route of administration, since astressin completely abrogate the effect of centrally administrated Ex-4 on the secretion of ACTH, but only slightly reduced ACTH to Ex-4 peripheral administration [140]. Moreover, GLP-1R knockdown in the PVN reduces HPA axis responses to acute and chronic stress [139].

### 3.2. GLP-1 Activates the Sympathetic Nervous System (SNS)

Using transgenic mice in which the glucagon promoter controls yellow fluorescent protein (YFP), varicose axons from PPG-neurons in central sites involved regulating autonomic functions have been observed [29]. The AP that densely express GLP-1 receptors could be one of the links for peripheral GLP-1 action to activate central autonomic regulatory sites since intravenous GLP-1 agonist (Ex-4) induce fos-IR in GLP-1-expressing neurons [29,141]. The NTS plays a crucial role in processing visceral afferent information and transmission to other nuclei in the brainstem, forebrain, and spinal cord [142]. In mice, medial NTS PPG-neurons receive monosynaptic input from vagal sensory neurons in the NG [24,143,144]. The chemogenetic activation of GLP-1R-positive vagal afferents induces Fos expression in this region [145]. Moreover, PPG-innervations are substantial direct to spinal sympathetic nuclei of the spinal cord that contain immunoreactivity for the enzymes that synthesize acetylcholine and nitric oxide. These neurons may regulate gut function but also affect cardiovascular control [39].

Moreover, YPF-PPG-neurons innervate ventral medullopontine catecholaminergic groups, including A1, A5, and C1 neurons, essential for regulating blood pressure and cardiovascular homeostasis [29,146,147]. Additionally, in the ventral medulla, serotonin (5-HT) neurons are widely innervated by YPF-PPG-neurons [29]. 5-HT-neurons regulate life-sustaining respiratory and thermoregulatory networks [148].

The sympathetic nervous system (SNS) innervates the adrenal cortex and influences plasma corticosterone production [149]. Sympathetic innervation of the gland modulates the diurnal rhythm in plasma corticosterone by increasing adrenal responsivity to ACTH and augmenting steroidogenesis [150]. In this sense, bilateral enucleation of rats and previous treatment with guanethidine attenuate the robust corticosterone response to Ex4, without affecting ACTH response, indicating the SNS’s role glucocorticoid effect of this GLP-1R agonist [140]. Moreover, Ex4 also increased the circulating levels of catecholamines by inducing the adrenal medulla [151].

Some of the firstly described effects of GLP-1R agonists were those affecting the SNS [121,151]. The activation of the GLP-1 receptor induces c-fos expression in neurons in autonomic control sites in the rat brain and the adrenal medulla, providing inputs to sympathetic preganglionic neurons [146]. Mice lacking the GLP-1R in the PVN show attenuated stress-induced cardiovascular responses accompanied by a decreased sympathetic drive to the heart [139]. GLP-1R activation by mechanisms dependent on the SNS can increase blood pressure and heart rate (HR) independently of the administration path (intravenous or intracerebroventricular) in rats [152,153] and in freely behaving and anesthetized mice [154,155]. Furthermore, a single injection of liraglutide or lixisenatide increases heart rate (HR) acutely in control and diminishes in *Glp1r*^CM-/-^mice (with selective cardiomyocyte disruption GLP-1R). This effect is abolished by co-administration of the β-adrenergic antagonists like propranolol or atenolol in freely moving and anesthetized mice [155,156] without further enhanced by attenuation of cholinergic signaling using the muscarinic receptor antagonist atropine [155,156]. Furthermore, the direct application of GLP-1 within the middle thoracic spinal cord’s subarachnoid space, a primary projection target of PPG-neurons, increases HR, suggesting direct activation of the SNS [155]. Other studies assume that the positive chronotropic effects of central GLP-1 receptor stimulation in mice result from depression of the heart rate’s parasympathetic modulation by inhibiting the neurotransmission to preganglionic parasympathetic cardiac vagal neurons [154].

Data in humans are conflictive since, although it has been demonstrated that GLP-1R agonist increases HR in healthy volunteers, obese subjects, and Type 2 diabetes mellitus (T2DM) patient, the role of the SNS it is not clear since some trials suggest an increase in SNS activity [157,158], but other studies found no effect [158,159].

Moreover, the activation of the SNS also appears to be related to pharmacological doses of Ex4 on blood glucose levels in rats, since acutely or centrally administered Ex4 induces hyperglycemia [151,160], independently of the insulinotropic and HPA activating actions. An effect mediated by GLP-1R [151] and abolished with the sympathetic blockade and adrenal medulloctomy [151]. 

The activation of thermogenesis in the brown adipose tissue (BAT) controls body weight homeostasis. In this regard, the SNS is essential for the control of BAT metabolism by the CNS [161]. GLP-1 emerges as a crucial modulator of BAT thermogenesis in mice by increasing SNS activity without altering peripheral insulin responsiveness [162]. Central administration of GLP-1R agonist induces BAT thermogenesis and facilitates adipocyte browning in white adipose tissue (WAT) through AMPK in the ventromedial hypothalamus (VMH) in rats [163]. Nevertheless, other hypothalamic areas are involved, such as DMH, since Glp1r knockdown increases body weight gain and adiposity, with a concomitant reduction in energy expenditure, BAT temperature, and uncoupling protein 1 (UCP1) expression [164].

### 3.3. GLP-1 Mediates Multiple Responses to Stressors

GLP-1 plays a critical role in the modulation of brain mechanisms regulating stress adaptation and mood. Many studies describe acute anxiogenic effects of GLP-1, which engage multiple structures to generate a coordinated response. The central administration of a GLP-1R antagonist blocks the plasma increase of ACTH and corticosterone induced by the EPM and decreases anxiety-like behaviors in the EPM, indicating that central nervous system (CNS) GLP-1 mediates anxiety responses [120]. As well, *Glp1r* knockdown in neurons expressing single-minded 1, a transcription factor abundantly expressed in the PVN in mice, reduced anxiety-like behavior [139]. However, chemogenetically activation of hindbrain GLP1 neurons shows no effect on anxiety-like behaviors, neither plasma corticosterone levels, showing the importance of hypothalamic GLP1R signaling for behavioral stress responses in mice [165]. However, the CeA, a brain region essential for the initiation of the stress response [166], appears critical for generating the anxiogenic effects of GLP-1 since the administration of the peptide in CeA does not modify plasma corticosterone levels but decreases the time spent in the open arms of the EMP [120]. Another neural substrate for GLP-1 control of anxiety-like behavior is the SuM. Selective activation of SuM, with Ex-4, decreases the time spent in the center of the open field arena in both male and female rats [37]. Otherwise, initiation of fear and sustained anxiety responses requires the recruitment of the BNST [137], knocking down the translation of GLP1-R mRNA in the anterolateral BNST in rats, decreases anxiety-like behavior in the open field test, including a loss of light-enhanced acoustic startle [52].

Moreover, the central administration of GLP-1 induces anxiety-like behavior in rats [138]. Also, central GLP-1 produces a proconflict effect in the punished drinking test while leaving activity and nociception measures unaffected, supporting an anxiogenic effect [167]. Besides, acute intraperitoneal, central or intra-dorsal raphe GLP-1 of or Ex-4 administration increases anxiety-like behavior using three different measuring tests in rats [117]. In contrast, chronic daily central treatment with the Ex-4 does not affect anxiety-like behavior but instead reduces depression-like behavior in the force swim test (FST) [117]. Contrarily, in humans, intravenously administered GLP-1 does not appear to have anxiogenic or panicogenic properties, even in patients with panic disorder [137]. 

Significantly, GLP-1 not just modulates the acute stress response, but can regulate HPA responsiveness to chronic stress. Exposure to chronic stress reduces PPG mRNA expression in a glucocorticoid-dependent manner, indicating that glucocorticoids produce long-term PPG downregulation and long-lasting reduction in PPG action [168], pointing out a role of GLP-1 in stress adaptation. Moreover, GLP-1 is involved in chronic stress-induced facilitation of corticosterone responses to a novel stressor, since the role of GLP-1 appears to be manifest following different stress exposure [169].

GLP-1 activity may amplify the effects of chronic stress on the organism. The i.c.v chronic administration decreases body weight in animals exposed to chronic stress, even though the GLP-1 administration itself does not precipitate chronic stress-like effects or long term HPA hyperactivity [169]. In contrast, sub-chronic Ex4 administration (subcutaneous bolus) produces several effects that resemble chronic stress. Overactivates, the HPA axis disrupts circadian glucocorticoid secretion, induces hypertrophy of the adrenal gland, decreases its sensitivity, impairs pituitary-adrenal stress responses induces reductions in both food intake and body weight [170]. Moreover, all those effects were abolished by adrenalectomy [140]. The regulation of the HPA axis by GLP-1 or Ex4 is independent of the metabolic state in rats [121]. In fasting, during which basal corticosterone levels are high, these peptides induce marked elevations of corticosterone levels, acting in conditions of metabolic stress, and independently of glycemic changes insulinotropic properties [121].

Challenges in the homeostasis induced by interoceptive stress activate central GLP-1 pathways [171]. The intraperitoneal treatment with the toxin lithium chloride (LiCl) activates c-Fos expression of GLP-1 neurons, including those with axonal projections to PVN in rats [172]. The administration of LiCl induces a pool of specific symptoms and behaviors in rats that have been used as indications of visceral illness [173]. Several of these responses also were caused by GLP-1, such as reduction of food intake [173,174] or conditioned taste aversion (CTA) [173,175]. The GLP-1R antagonist blocks the effect of LiCl to reduce food intake, induces pica, and produces a CTA in rats [173]. Like the rat, LiCl activates PPG-neurons, induces anorexia, and CTA formation in wild-type mice, but LiCl does not evoke aversive effects in mice lacking GLP-1Rs, indicating species differences [176]. GLP-1R activation in the CeA appears to mediate some of the responses to peripheral illness, CeA GLP-1 infusion, but not the inactive GLP-1(9–36), results in a strong CTA, without inducing anorexia [177]. Furthermore, intra-amygdala administration of des-His1, Glu9-exendin-4, the GLP-1R antagonist, prevents taste aversion learning in response to i.p. injections of the LiCl [177].

Moreover, gastric distension stimulates vagal mechanoreceptors, predominantly located in the proximal and distal stomach, and lastly, increases c-Fos levels in NTS neurons expressing GLP-1 and GLP-2 in rats [178]. Furthermore, central GLP-1 is a physiological modulator of stress-induced colonic motility in the rat, since centrally but not peripherally administered GLP-1 increase fecal output after immobilization stress, an effect reverted by exendin (9–39), a competitive antagonist at GLP-1R [179]. 

## 4. GLP-1 in the Control of Food Intake. Crosstalk with the Stress System

The effects of GLP-1 on food intake have been of considerable attention in clinical and basic studies and described in different species, including rats [74], mice [12], or humans [180]. Peripherally, intestinal GLP-1 acts as a shorter-term prandial satiation signal [181,182,183], it is secreted in the response of food ingestion [17], reduces meal size in rats and humans [181,184], and increases intermeal intervals, accounting for its suppressive effect on food intake [181]. Moreover, GLP-1R blocking with Ex9 increases food intake in rats [181]. The peripheral administration of native GLP-1 requires a postprandial state to express biological activity to inhibit food intake [185]. Accordingly, oral, but not intra-3rd-ventricular (i3vt) or IP glucose potentiated GLP-1s anorectic action [186]. The physiological doses of GLP-1 that inhibit eating do not induce avoidance in rats [187] or gastrointestinal malaise in humans [184].

GLP-1′s peripheral effects on food intake point towards a role for vagal afferents by the vagus nerve. The vagal afferents neurons (VANs) of the NG express GLP-1R [188] and innervate the gastrointestinal tract, liver, and portal vein [189]. Endogenous GLP-1 acts in a paracrine fashion to stimulate adjacent GLP-1R on the dendritic terminals of the celiac and gastric branches of VANs that innervate the gut, reducing food intake via vagal-NTS glutamatergic signaling and also mediate insulin release via vago-vagal reflex [190]. Knocking down GLP-1Rs in VANs by injecting a lentiviral vector in the NG increases meal size, accelerates gastric emptying, increases postmeal glycemia, and blunts insulin release [191]. Also, subdiaphragmatic vagotomy reduces the anorexic effect of peripherally GLP-1 administration [192]. Indeed, GLP-1R on VANs is modulated by feeding, since GLP-1Rs expressed on vagal afferent neurons are trafficked to the membrane in response to a meal in 18 h fasted and then re-feed rats, giving a possible explanation of the observation exogenously administered GLP-1 only inhibits food intake after feeding [185]. All were together, suggesting that GLP-1 receptors in VANs contributed to the incretin-linked effects after a meal [191]. However, there is also evidence showing that other additional mechanisms may be involved in GLP-1 peripheral effects contributing to food intake reduction. Thus in the rat [193] or mice with visceral nerve-specific deletion of GLP-1R [194], the vagotomy does not modify food intake. Moreover, subdiaphragmatic vagal afferent deafferentation does not prevent lowering food-intake after long-lasting liraglutide treatment [195]. In this regard, the activation of areas outside the blood-brain barrier (BBB) could have relevant effects on eating, since peripherally injected ^125^I-labeled GLP-1 binds to the subfornical organ and the AP, which both have close neuroanatomical connections with hypothalamic areas involved in water and appetite homeostasis [196]. Besides, GLP-1R agonist reaches regions shielded by the BBB such as arcuate, PVN, or SOP nuclei of which most were intersected by projections from neurons in the lateral PBN. [197]. For example, liraglutide can access the brain either by diffusion from the circumventricular organs (CVOs) or by uptake through specialized cell structures intercepting the BBB protected brain regions with the CVOs [197], indicating that neuronal targets of applied pharmacological GLP-1R agonism can differ from engaged by NTS-derived GLP-1. Moreover, exogenous administration of GLP-1R agonists failed to reduce food intake (and gastric emptying) in *Glp1r*^ΔWnt1-^/^-^ mice or *Glp1r* conditional KO mice, reflecting the importance of neural GLP-1R populations for the pharmacological control of GLP-1-regulated feeding [194,198].

Rodent PPG-neurons are sensitive to satiety peripheral signals, including leptin [24] or cholecystokinin (CCK) in fed state [199,200]; also, gastric distension increases c-Fos-expression in NTS neurons [178] but is unaffected by GLP-1, PYY, or ghrelin [24], bringing out the potential role of these neurons as central integrators of several satiety signals in the NTS. In this regard, PPG-neurons do not express GLP-1R but receive direct synaptic input from sensory vagal neurons [24] and express functional leptin receptors [24,201]. However, the effects differ between species since PPG neurons in mice but not in rats, are responsive to leptin [202]. Leptin systemically administered elicits robust pSTAT3-ir within the NTS, but outside of GLP-1 neurons in rats [203]. It seems that in mice, PPG cells are mostly second-order neurons, receiving direct input from vagal afferent fibers [24]. Recently it has been described that GLP1 cells in the NTS represent a subset of LepRb^NTS^ cells in mice, while in NTS rats, GLP-1- and LepRb-containing cells are distinct [204]. In mice, the activation of LepRb^NTS^ neurons mediates a robust and durable suppression of food intake independently of GLP-1 signaling [204], bringing out NTS PPG system differences between species

Centrally administrated GLP-1 reduces food intake in fasted rats [205] and reduces water intake [174]. The effect induced by the central administration of the GLP-1 or the GLP-1 analog, liraglutide is short-lasting compared to large doses of CCK, just observed on the first day of treatment [163,206], and remission at 48 h [163,207]. Similarly, the GLP-1R agonist’s peripheral administration, Ex4, reduces food intake in 24 h-fasted rats [151] and after the onset of the dark phase, reducing meal size and increasing intermeal interval [181]. Conversely, blocking endogenous GLP-1R with Exendin 9-39 (Ex9) increases rats’ food intake [181]. However, disruption of GLP1/GLP1R signaling in the central nervous system is not associated with perturbation of feeding behavior or obesity in mice, showing species-specific differences [208]. The dose, pharmacokinetics, or the antagonist’s selectivity may be implicated in the discrepancies observed between studies [12].

GLP-1R activation, in rat, increases AMAPA/Kainate-mediated glutamatergic signaling in the NAc and VTA. This activity is, at least in part, responsible for reduced food intake and weight loss GLP-1-induced [209,210]. Glutamate is considered the primary excitatory neurotransmitter in the CNS by exerting depolarization in the postsynaptic neurons. It has been suggested that Ex-4 improves the levels of glutamatergic receptors (GluN1) and transporters (EAAT-2) [211,212] hence improving glutamatergic tone. The involvement of AMPA/Kainate receptors in response to GLP-1 at the central level has been further confirmed [213]. It has also been reported an increment of GLT-1 (primary glutamate transporter in the hippocampus) mRNA in mice treated with Ex-4 [211]. Similar results are found in cultured primary astrocytes [212] and hypothalamic slices [96]. On the other hand, in vivo, liraglutide induces weight loss and reduction in food intake [214], and it has been proposed that the glutamatergic hypothalamic neurons were required for the liraglutide-induced effect [215].

PPG-neurons in the NTS respond to abroad array of interoceptive signals that can suppress food intake, including hormonal, thermal, osmotic, gastrointestinal, cardiovascular, respiratory, and inflammatory signals in both rats and mice, bringing out the role of GLP-1 PPG-neurons in modulating food intake in response to mainly intense or stressful stimuli [13,48,122,216,217,218,219]. 

In this context, it is clear that GLP-1 released from the gastrointestinal tract after a meal plays a critical physiological role in satiety [181]. However, GLP-1 from NTS-PPG-neurons projecting throughout the brain to many hypothalamic areas emerges as a critical modulator involved in controlling energy homeostasis and reward [22,207]. Moreover, cells from these areas project to nuclei associated with reward and motivation [47]. In this context, GLP-1 could act as a coordination link between homeostatic and hedonic pathways in the control of food intake preferentially in response to stress (Figure 1).

### 4.1. GLP-1 Anorectic Action and HPA Activation

The neural circuits that regulate energy intake converge on the PVN, in which there are CRF containing neurons implicated in the regulation of the HPA axis, thereby providing overlap between the stress and feeding systems [220]. The amount and type of food eaten can be influenced by stress. Many types of stress are associated with reduced food intake [221]. However, although 20% of people do not change eating behaviors during stress periods, it has been reported an increase in total calories eating under stress experiences [2,222], emphasizing eating a more significant proportion of calories from highly–palatable foods [223]. Under these conditions, chronically stressed individuals are more susceptible to weight gain, obesity, type II diabetes, or cardiovascular diseases [2].

The initial component of the HPA axis, the CRF, is located in neurons in the PVN of the hypothalamus, but also in extra-hypothalamic limbic structures like the extended amygdala (included the BNST and the CeA), as well as in hindbrain structures like the Barrington´s nucleus adjacent to the locus coeruleus and dorsal raphe nuclei [224,225], playing a role in addiction and several psychiatric disorders [224,226]. This peptide acts through two significant receptors; the CRF1 activation is associated with increased stress responsiveness, while CRF2 receptor activation suppresses food intake and decreases stress responsiveness [227]. There is increasing evidence that CRF neuronal activation in the PVN by GLP-1 likely contributes to food intake suppression. Using optogenetics and chemogenenetic approaches, and slice physiology, it has been described that 50% of CRF-neurons receive direct projections from NTS PPG-neurons and exist a direct synaptic connection between these neurons. Moreover, GLP-1R activation increases the excitatory synaptic strength to CRF-neurons in mice, through enhancement of AMPA receptor subunit membrane trafficking and the inhibition of CRF neuronal activity blocks GLP-1 induced satiety in the PVN. [213]. CRF also mediates the anorexic effect of GLP-1 in chicks and mediates the inhibition of gastric emptying induced by GLP-1 in rats [228].

At several concentrations, such as pico- and nano-mol/L, both GLP-1, and Ex4 are capable of enhancing a transient Cl^-^ inward current in neurons from different brain areas, including the hypothalamus and the hippocampus [97,229,230]. In this regard, GABA mainly mediates the inhibitory transmission, and Ex-4 applied in the micromolar range increases hippocampal inhibitory transmission by activating GABAA receptors at pre-, post- and extra-synaptic sites [230,231]. Besides, the addition of GLP-1 to rat hippocampal slices results in GABA release, and this effect is abolished in the presence of bicuculline (a GABAA receptor antagonist) [230]. Interestingly, dipeptidyl peptidase-4 (DPP4) inhibitors enhance the endogenous GLP-1 levels and enhance GABAergic transmission in rat hippocampal neurons in vivo [232]. In the same way, in experiments using microdialysis, it has been verified that the application of GLP-1 increases the GABA concentration in the rat PVN [229]. 

Besides, GLP-1 mediates illness or stress-induced anorexia. The previous injection of GLP-1R antagonist blunts the potent LiCl-related suppression of food intake, inducing pica, and producing CTA [172,173]. This effect supports the functional role of endogenous GLP-1-containing neural pathways as mediators of aversive behaviors in rats [176]. Also, the central administration of the GLP-1R antagonist (exendin 9-39) reduces in a 60% the anorexic response to lipopolysaccharide (LPS) [233]. Likewise, rats’ metabolic state is an essential factor contributing to acute stress, since caloric restriction reduces HPA axis activity [234] with lower baseline and stress-evoked plasma ACTH levels [235]. In this regard, acute restraint stress suppresses dark-onset food intake in rats fed ad libitum. This effect is reverted by the central infusion of GLP-1R antagonists, which blocks restraint stress-induced hypophagia and reduces anxiety-like behavior. These data suggest an underlying mechanism by which short-term negative energy balance attenuates neuroendocrine and behavioral responses to acute stress that involves GLP-1 signaling [122].

Furthermore, dexamethasone suppression of the HPA axis augments the Ex4 induced anorexia [149], illustrating Ex-4 and dexamethasone’s synergistic effects co-administration. In other brain regions, such as the BNST, the center of integration for limbic information regulates the affective and physiological components of anxiety [137], GLP-1R blockade attenuates stress-induced hypophagia in mice [33]. In this region, the GLP-1R mRNA is expressed by a subset of GABAergic neurons, and some of these GLP-1R-expressing neurons also express CRF mRNA [28].

### 4.2. GLP-1 Anorectic Effect: Hypothalamic Actions 

Subcutaneous administration of native the GLP-1 induces c-fos activation in the hypothalamus and hindbrain [236], indicating that peripheral administration can activate central circuits. The peripheral administration of liraglutide labeled with a fluorescent probe was observed in hypothalamic regions protected by BBB, such as the arcuate, the PVN, the SON, and supraoptic decussation [195].

Likewise, activation of central GLP-1R by icv or i.p. administration of GLP-1 or its analogs induces satiation [195,237]. In the brain, the PVN contains different neuronal populations related to appetite regulation, stress response, and other neuroendocrine functions [132,238], one of the neuronal populations linked to satiety signaling express GLP-1 receptors [132]. By fiber photometry, it has been described that food discovery modulates this neuronal population in an anticipatory manner, and its stimulation orchestrates feeding behavior [132]. Moreover, the postnatal ablation of PVN GLP-1R causes increased food intake, body weight gain, and obesity [213] (Figure 2).

Also, in the hypothalamus, the administration of liraglutide in the ARC reduces food intake 24 h after injection [163]. In the ARC, numerous cells express GLP-1R mRNA, and many of them also co-express proopiomelanocortin (POMC) neurons mRNA. Approximately an average of 68% of POMC-neurons was found predominantly in the ARC mediolateral part, co-express GLP-1R mRNA, and about half of the GLP-1R-expressing cells, preferably in the caudal part of ARC, do not express either POMC or neuropeptide Y (NPY) mRNA [240]. Furthermore, liraglutide peripherally injected targets GLP-1Rs located in the ARC, and it is internalized by POMC/transcript regulated by cocaine and amphetamine (CART) neurons, adjusting the neuronal activity [195]. This effect was observed in brain slices from Pomc-EGFP mice after stimulation with GLP-1(7-36)amide, that dose-dependently depolarizes POMC-neurons and increases the frequency of action potentials [195]. Interestingly, GLP-1(7-36)amide stimulation increased the frequency of GABAergic currents onto POMC-neurons, suggesting that GLP-1 activates POMC/CART-neurons directly at the level of the cell body and that the NPY/agouti-related peptide(AgRP) pathway is inhibited at the NPY/AgRP neurons via GABAergic interneurons [195].

Other hypothalamic nuclei are innervated by hindbrain GLP-1 neurons and express GLP-1 receptors [39,71] such as the LH, involved in motivation feeding behavior [241]. In this regard, specific activation of GLP-1R in this region decreases food intake [163], the intra-LH microinjection of Ex-4 reduces food-motivated behavior, and knocking down the GLP-1R increases food reinforcement and body weight [242]. The microinjection of GLP-1 into the VMH and DMH reduces food intake by 30% and 48%, respectively, after 20 min of administration in fasted rats [243]. In fed rats, specific activation of the GLP-1R by liraglutide in the ARC, LHA, and PVH decreased rats’ food intake and body weight [163]. In contrast, no effects were described when liraglutide was injected in the DMH; however, the treatment in the VMH decreases body weight in food intake–independent manner related to increased thermogenesis in the brown adipose tissue in rats [163]. Also, pharmacological activation of the GLP-1R in the VMH by Ex-4 reduces food intake, activating mTOR signaling, indicating that glucose metabolism and inhibition of AMPK are both required for this effect [244]. In contrast to the clear acute pharmacological impact on food intake, knockdown of the VMH *Glp1r* conferred no changes in energy balance in either chow- or high-fat-diet-fed mice, glucose homeostasis, or the response to peripherally administered GLP-1R agonist [244,245], suggesting therefore that classic homeostatic control regions are sufficient but not individually necessary for the effects of GLP-1R on nutrient homeostasis [244,245].

### 4.3. GLP-1 Anorectic Effect: Forebrain/Hindbrain/Mesolimbic Actions 

Beyond the hypothalamus, other brain areas, such as the paraventricular thalamic nucleus (PVT), the PBN, VTA, medial prefrontal cortex, amygdala, NAc, or hippocampus, involved in the control of food intake presents GLP-1 signaling [190]. Many of these sites are activated by food pleasures but not all brain activations that code food pleasure necessarily causes or generate the pleasure, since other brain activations are likely to be secondary or consequent to the pleasure, and in turn, could cause motivation, learning, cognition or other functions [246].

The PBN contains several subpopulations of neurons that regulate taste [239], integrate neural signals associated with satiety from neuronal populations on the PVN, and receive inhibitory projections from AgRP neurons [247]. PBN receives excitatory glutamate signaling from NTS neurons’ subpopulation responsible for integrating visceral and gustatory inputs [248]. GLP-1 producing neurons from the NTS projects to the lateral PBN [35,249] and local activation with Ex-4 inhibits food intake of chow and palatable food, the motivation to work for palatable food, and decreases body weight gain [35,249], thereby implicating this brain region in the hedonic aspect of feeding [190]. Moreover, Ex-4 increases neuronal firing, and the expression of calcitonin gene-related peptide (CGRP) in this nucleus resulted in anorexia [249]. This anorectic effect is not related to nausea/malaise since Ex-4 does not induce pica response [35].

The PVT neurons receive projections from hindbrain regions and substantial inputs from the hypothalamus and project to forebrain sites such as the core and shell of the NAc, involved in reward and motivation function [250]. Neurons in PVT express GLP-1R and receive monosynaptic inputs from NTS preproglucagon neurons activated by food intake [23,36]. PVT GLP-1R agonism by intra-injections of Ex-4, reduces food intake, food-motivation, and food-seeking, while blocking GLP-1R signaling with PVT Ex-9 injection, increases meal size and food intake [36]. Moreover, PVT cells that express the GLP-1R project to the NAc and electrophysiological results reveal that PVT GLP-1R signaling reduces PVT-to NAc projecting neurons’ excitability, contributing to the motivational aspects of feeding control [36].

Reward-related regions such as VTA and the NAc, the core and the shell subregion, receive NTS GLP-1 neuronal projections [22,31,251]. Pharmacological treatment with Ex-4 delivered to the VTA in overnight food-deprived rats reduces one-hour sucrose intake, 24 h chow intake, and 24 h body weight. Also, Ex-4 treatment in the VTA, NAc core and NAc shell can reduce high-fat diet intake in not food-deprived rats [31]. However, if the animals are maintained on chow, Ex-4 injections into the VTA, NAc core, or shell does not suppress food intake, suggesting a role of GLP-1 signaling in motivation to feed [31]. In this regard, Ex-4 reduces food-reward behavior in the conditioned place preference and progressive ratio operant-conditioning, since peripheral Ex-4 treatment of rats blocks preference for chocolate pellets and decrease motivated behavior for sucrose [252]. Moreover, post dark onset intra-NAc core treatment with GLP-1 also reduces food intake 24 h after injection without affecting body weight. However, no effect was described after NAc-shell injection [251]. Furthermore, the NAc core injection of Ex-9 increases food intake two hours after posttreatment [251]. This negative energy balance induces by GLP-1R activation in NAc is, in part, thought a glutamatergic, AMPA/kainite receptor-mediated mechanism [210]. 

However, manipulations related to NAc affect reward-motivated behavior and can affect feeding by changes in food palatability [207]. Blocking NAc GLP-1R signaling with Exe9 enhances sucrose solutions’ palatability in meal patterns and microstructure studies of ingestive behavior in rats [253]. Also, NAc Ex9 did not affect licking for nonnutritive saccharin (0.1%), suggesting that the presence of nutrients in the gut may be required for endogenous stimulation of NAc [253]. So, at this site, GLP-1 R stimulation reduces the reward associated with food palatability and contributes to satiety reducing meal size, with no effect on meal frequency when rats consume sweetened condensed milk or sucrose [207,253].

In this regard, GABA neurons in NAc regulate homeostasis, especially feeding behavior [254]. GABAergic medium spiny neurons transmit signals to the ventral pallidum and susbstantia nigra of the basal ganglia after NAc has integrated the information, regulating motivation-related behavior. So neural projections of these cells from the zona incerta act on NAc to enhance gastric function and food intake via GLP-1R signaling as an essential effector [255].

Moreover, the SuM nestles between the LH and the VTA, which regulates ingestive and motivated behavior, express GLP-1R [74]. Knocking down the GLP-1R in this nucleus increases food-seeking and adiposity in obese male rats without altering food intake, body weight, or food motivation in lean or obese females [74] and induces anxiolytic responses in female rats [37].

Another brain region that is important to take into consideration related to food intake is the hippocampus. This forebrain structure is neuroanatomically interconnected with several regions before mentioned, such as the PBN, VTA, NAc, or amygdala [256]. Besides, hippocampal neurons integrate learned experience with the external and internal context to influence decisions about when, where, what, and how much to eat [256]. It has recently been described that gastrointestinal-derived vagal sensory signaling endogenously promotes hippocampal-dependent learning and memory function in rats [257]. Endocrine pathways involve the interaction between different satiation signals and the hippocampus, such as leptin [257], ghrelin [258,259], and also GLP-1 [260]. The GLP-1R is mostly expressed in neurons in ventral CA1 and CA3 pyramidal layers [23]. The activation of GLP-1R by Ex-4 in the ventral hippocampal formation induces hypophagia and reduces body weight in rats, through a specific reduction in meal size with no effect on meal frequency by mechanisms other than nausea [260]. Furthermore, this HPFv GLP-1 signaling is relevant for feeding since the administration of Ex (9-39) increased food intake by 30%, six hours after its administration [260]. In this regard, a novel hippocampus-hypothalamus-hindbrain pathway regulating meal size control has been described in which ventral hippocampus ghrelin signaling counteracts the food intake–reducing effects produced by the GLP-1R agonist, Ex-4, and also other gut-derived satiation signals, including CCK, amylin, and mechanical distension via downstream orexin signaling to the hindbrain laterodorsal tegmental nucleus [259]. 

### 4.4. Stress in Obesity: GLP-1 and the Motivation to Feed

It has been shown that chronic stress, mild hypercortisolemia, and prolonged SNS activation contribute to the clinical presentation of visceral obesity and type 2 diabetes [228]. Considerable evidence demonstrates that feeding behavior is influenced by stress, and this relationship also seems to be strongest among individuals who are overweight and those who binge eat [1,261]. In humans and animals, a shift toward choosing more pleasurable or palatable calories occurs whether or not total caloric intake increases with stress [2].

Metabolic systems and brain reward systems play a significant role in feeding behavior [5]. Similar brain regions are activated by palatable food in the rat and humans, such as the dorsal and ventral striatum, VTA, LH, NAc, CeA, and basolateral nuclei of the amygdala, the hippocampus, and reward-related cortical structures, as well as the neurotransmitter system (dopamine, serotonin, opioids, and endocannabinoids) [4,5]. Many foods of intense sweetness and fat are potent rewards [3], promote eating, and trigger learned associations between the stimulus and the reward (conditioning) [4], but also induce changes in carbohydrate and fat metabolism, insulin sensitivity, and appetite hormones, increasing salience and motivation for food intake that may alter energy homeostasis [262]. Overconsumption of high palatable foods reduces reward thresholds along with an upregulation of extrahypothalamic CRF in the amygdala and limbic striatal pathways involved in the regulation of stress response [262,263]. It may also promote food craving, increasing overeating risk, and stress-induced high palatable food-seeking [262,263]. The loss of control and overeating results in compulsive eating behavior, certain forms of obesity, and the recently proposed term of “food addiction” [264]. Recently, it has been demonstrated that binge-like eating over several weeks in a model of intermittent fat feeding in rats affects the GLP-1 system, decreasing PPG mRNA expression in the NTS, suggesting downregulation of central GLP-1 signaling [265], accordingly with dysfunction in satiation processes that generally serve to limit food intake observed in different animal models [265]. Moreover, rats with intermittent access to a high-fat diet show large meals [266] and reduced responsiveness to intragastric nutrients and amylin, supporting the idea of reduced sensitivity to satiation signals under these conditions [266]. A possible loss of sensitivity to GLP-1 is not yet proven and could also explain the lack of inhibition of large meals observed in this animal model.

Moreover, there seems to exist a strong link between obesity and impaired function of the reward network. In rodents, extended access to palatable food induces weight gain and a worsening brain reward deficit, characterized by a decrease in reward responsiveness in the LH [5]. Furthermore, feeding is associated with dopamine release in the dorsal striatum [267], and it has been described in obese individuals and obese rats, a decreased expression of striatal dopamine-receptor D2R [5,268], similar to those found in patients under drug addiction [269]. Interestingly, knocking down striatal D2R increases the emergence of compulsive-like eating in rats with access to palatable food [268], and this compulsive behavior continues in the presence of an aversive conditioned stimulus pointing out also the role of CeA [270]. Amygdala D2 dopamine-receptor activation reduces food intake and operant behavior for sucrose, whereas D2 receptor blockade increased food intake, reducing operant behavior [271]. However, the amygdala plays also a role in stress-related hedonic eating, in which a decreased expression of CRF was observed under a palatable diet, and withdrawal from such a diet can produce a heightened emotional state and maladaptive coping responses that increase the drive to obtain palatable food as a rewarding source in an aversive environment [272].

Different studies in diet-induced obesity mice and obese humans suggest central resistance to different metabolic hormones that control food intake, such as leptin or GLP-1 [273]. Gut-brain communication is altered by high-fat consumption [274], and impairs the anorectic response to Ex-4 [275], altering the anorectic response to peripheral administration of GLP-1R agonists, delaying the onset but also prolonged the action on the depression of food intake [276]. Moreover, GLP-1 deficiency may play a critical role in developing the pathophysiology of obesity, since this hormone decreases food intake and body weight, as was described previously in this review. Furthermore, in monogenic obesity (1% of total cases of obesity), the PCSK1 gene encoding the PC1/3 enzyme losses its function, or it is mutated [277]. However, obesity-related to environmental and societal changes are related to this gene since single-nucleotide polymorphisms at three loci of PCK1 are related to an increased risk of obesity [277]. In this regard, a reduction in GLP-1 secretion has been described in obesity with altered L-cell responsiveness to carbohydrates [278], accompanied by insulin resistance [279]. Ghrelin and leptin are potent modulators of GLP-1 secretion by L-cells [280,281]; both hormone systems are impaired occurring secondary to obesity [282,283], causing functional deficits in GLP-1 signaling [277]. Moreover, the incretin effect of GLP-1 is altered in obesity [271], and it is inversely correlated with BMI [284]. Furthermore, in obesity, an accelerated gastric emptying could be observed that could be related to reduced GLP-1 signal, which could predispose to an earlier onset of the next meal, contributing to overeating [285]. 

Using functional MRI (fMRI), it has been demonstrated that brain regions involved in reward processing are altered in obese individuals [286]. The acute treatment with a single dose of the GLP-1 receptor agonist, exenatide, reduces brain responses to food cues in normoglycemic obese and T2DM patients, also correlated with reductions in food intake, but without effect in weight loss [286]. Moreover, these effects are blocked by exendin 9-39 [287]. After ten days of treatment with liraglutide, an increase in the activation of the right insula and caudate nucleus related to chocolate milk was observed in obese T2DM patients compared to lean individuals. This effect was not achieved after 12 weeks when apparent effects on body weight were observed [288]. Also, the administration of liraglutide at the dose approved for obesity treatment reduces body weight at five weeks but does not show differential activations in response to food cues. However, with control for the change in body weight, an increase in orbitofrontal cortex activation was achieved, indicating the beginning of counter-regulatory changes in response to weight loss [289], which could be related to the eventual weight-loss plateau observed with this and other weight-loss medication.

## 5. Conclusions

Acute or chronic exposure to stress evokes different physiological and behavioral responses that considerably alter metabolic and behavioral status in humans and experimental animals [223]. The activation of the HPA axis and SNS, in response to stress, increases glucocorticoid and catecholamines synthesis, facilitating glucose availability to fuel the metabolic demands of other physiological and behavioral stress responses [125,126,222,290]. Glucocorticoids regulate body fat accumulation and increase appetite, food intake, and body weight gain [290,291]. Chronic stress and obesity are closely related to disordered eating syndromes, including bingeing or night predominant intake [223].

In the brain, GLP-1 acts as a neuromodulator. Produced by PPG-neurons in the NTS [21], modulate and process peripheral and central signals to maintain the homeostasis. PPG-neurons innervate numerous brain regions responsible for modulating many physiological functions such as metabolism, reward-seeking behavior, and stress response. This neuropeptide modulates the stress response, driving the HPA axis, and regulating the SNS and mediating the limbic system eliciting responses to homeostatic and psychogenic stressors [29,47,120,121,292]. Moreover, GLP-1 controls food intake, modulating the energetic balance, acting on GLP-1R in a multitude of energy balance-relevant nuclei in the hypothalamus [21,205], but also in other mesolimbic brain areas involved in reward [23].

In conclusion, GLP-1 in the brain appears as a neuromodulator that coordinate food intake in response to the physiological and stress-related stimulus, since GLP-1 signaling affects different brain areas that control diverse aspects of feeding to maintain homeostasis. However, when this equilibrium is disrupted (obesity, diabetes, chronic stress [168,285]), the GLP-1 signaling is dampened, and dysregulation in eating behavior may appear. Future studies should address how satiety activities, observed under HPA axis activation, depend directly or indirectly on GLP-1 since it is known that CRF has potent anorexic effects [293] and how the nature and duration of the stressors affect food intake and behavior involving GLP-1. It is still unknown how stress influences central GLP-1R signaling in metabolic pathologies, such as obesity or diabetes. Neither how the HPA axis and GLP-1 interact with the control of feeding under these metabolic diseases. Answer these questions that will open new relevant pharmacological actions of GLP-1 analogs in the control of obesity.

## Figures and Tables

**Figure 1 nutrients-12-03304-f001:**
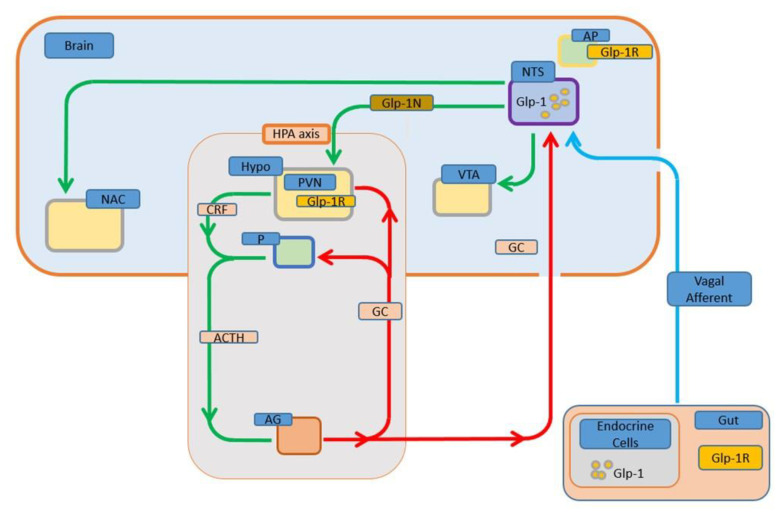
Representative scheme of the interactions and effects of GLP-1 at the central level. NAC:Nucleus Accumbens; CRF: corticotropin-releasing factor: ACTH: corticotrophin; HYPO: hypothalamus; P: pituitary; AG:adrenal gland; PVN: paraventricular nucleus; GLP-1R: glucagon-like peptide 1 receptor; GC: glucocorticoid; GLP-1N: glucagon-like peptide 1 neurons; VTA: ventral tegmental area; NTS: nucleus of solitary tract; AP: area postrema.

**Figure 2 nutrients-12-03304-f002:**
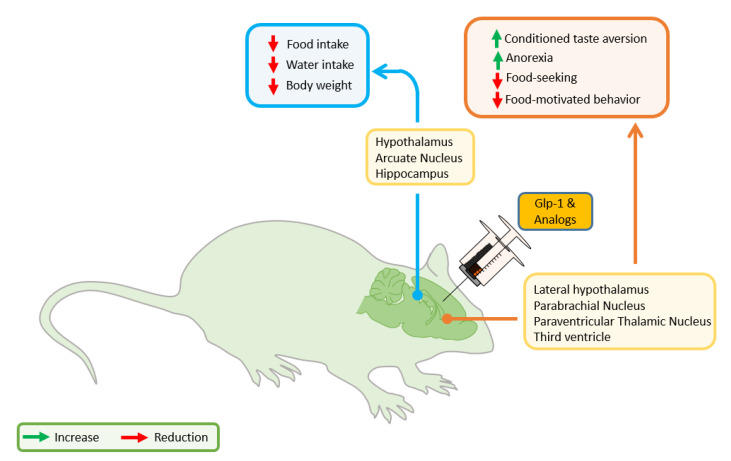
The figure represents the effects of the central administration of GLP-1 and analogs on intake and some related behaviors schematically. The central administration of GLP-1 or analogs produces several physiological effects such as decreased food intake [158,206,230], water intake [171], and body weight [239]. Likewise, it also produces more related to food behavior effects, including increased CTA [169,170] and anorexia [220]. Besides some effects such the decreased of food-seeking [21,34,201,224] and food-motivated behavior [21,34].

**Table 1 nutrients-12-03304-t001:** Electrophysiological effects of GLP1 in the hypothalamus.

Compound	Effect	Channel/ Current	Preparation	Dose	Tissue	Model
Ex-4	activation	*I* _Nav_	in vitro	1 µM	HPN	mouse
Ex-4	activation	nonselective cationic	in vitro	1 µM	HPN	mouse
Ex-4	non effect	*I* _A_	in vitro	1 µM	HPN	mouse
GLP-1 and analogs	¿blockade?	K2P (TREK) ^1^				
Ex-4	depolarizing		in vitro	1 µM	HPN	mouse
Ex-4	reduction	*I* _Cav_	in vitro	1 µM	HPN	mouse
Ex-4	reduction	*I* _AHP_		1 µM	HPN	mouse

Ex-4: exendin-4; K2P: two-pore domain potassium channels; I_A_: Potassium current A-Type; I_AHP_: afterhyperpolarization current; I_Cav_: Voltage-dependent calcium current; HPN: Hypocretin Neurons; ^1^ Tentative Hypothesis. Adapted from [96].

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
