# Peer review of "Glucagon-Like Peptide-1 (GLP-1) in the Integration of Neural and Endocrine Responses to Stress"

_nutrients, 2020, doi:10.3390/nu12113304_

Round 1

Reviewer 1 Report

A substantial amount of work has clearly gone into it, and I think there is a valuable and informative paper 'in there' somewhere. However, despite the extensive feedback given during the first round of review (particularly the very generous list of suggestions for additional citations provided by the other reviewer), there are at least as many factual, conceptual and presentational errors in the revised version.

Author Response

We thank the reviewer for his/her comments. We would like this latest version to be the referee's liking, and we thank the referee for all the help have given us to improve this manuscript

Reviewer 2 Report

The authors of ‘Glucagon-like Peptide-1 (GLP-1) in the integration of neural and endocrine responses to stress’ provide a summary and discussion of the role of GLP-1 in stress. I am happy to say the revised version has addressed most of my concerns from the initial review. However, a few remain unresolved and a number of new concerns have appeared following addition of new paragraphs. I hope the authors will find my comments constructive.

Broad comments:

  • The flow and logic of the manuscript is still somewhat confusing. Future readers would benefit from a more logical discussion of what the individual paragraphs tell us about the topic of the review: GLP-1 in stress.
  • Fig 1 remains unclear and inaccurate to this reader. While the authors added an arrow indicating vagal input to the NTS, the figure still appears to suggest that vagal afferents transmit GLP1 peripheral GLP-1 signalling to the NTS via the AP. Can the authors provide references to the literature this is based on?
  • In addition, the authors state in the figure legend that the schematic shows effects of GLP-1 peripherally. That is mostly not the case (bar the effect on sensory vagal input), so I suggest the authors rephrase the description of the figure.
  • Abbreviations are used inconsistently. As an example, SNS is used to denote ‘sympathetic nervous system’, yet the full term ‘sympathetic nervous system’ appears eight times in the text. Please check for this type of error throughout the manuscript.
  • The revised version should be carefully edited to remove grammatical/typographical errors, missing or duplicated words, etc.
  • I found at least one example of a duplicated reference (133 and 135). Please double check the bibliography.

Specific comments that were incompletely addressed in the revision (line numbers refer to the previous version):

  • P1L38: I do not like the term ‘the homeostatic system’ in this place. The homeostatic system cannot act on the hypothalamic and brainstem circuits. If anything, those circuits are part of ‘the homeostatic system’. Homeostasis is a state of equilibrium, not a system like the sympathetic nervous system or the central GLP-1 system.

‘The homeostatic mechanism’ is also not a correct term. I would completely rephrase this sentence to avoid confusion.

  • P2L59: ‘The neurons of the NTS mainly project efferents to the PVN in the hypothalamus [14], providing glutamatergic input.’ It is correct that the NTS provides massive input to the PVN, catecholaminergic as well as glutamatergic! However, the projections are from the NTS are widespread! E.g. the PBN is a dense target too. In addition, it would seem more relevant to discuss the widespread projection patterns of GCG neurons here (Llewellyn-Smith et al., 2011, 2013, 2015).

This comment was not addressed.

  • P6L243: This paragraph is very confusing. Are the authors saying that GABA induces release of GLP-1 from L cells? And that beta-cells release GABA in response to GLP-1? Please rephrase and explain the relevance to stress.

This is still very confusing and has not been addressed. I fail to see how the two statements are related. One is talking about the effect of GABA on GLP-1-producing cells. The next about the effect of GLP-1 on GABA release. And all of this in the periphery. Can the authors make these statements clearer and indicate their relevance to the topic of the manuscript?

  • P9L364: Compared to other hormones (e.g. CCK) a day is actually quite a long-lasting effect.

This is still incorrectly described in the text (L455). Unless large doses of CCK are used (and these induce visceral illness) then the effect of CCK on food intake is very brief (only clear at 30mins post-injection). The effect of liraglutide is comparatively long.

  • P11L430: How do the authors explain this discrepancy between liraglutide and GLP-1?

The authors did not address my concern. Now L592.

  • P12L512: CRF is actually expressed in Barrington’s nucleus adjacent to the locus coeruleus. Only very little is present in LC. See e.g. (Holt et al., 2019; Peng et al., 2017).

This was not corrected in the text, which still states that CRF is expressed in locus coeruleus. Please change.  

  • P12L526-527: This is only true for rats. (Lachey et al., 2005).

While the authors do include the information regarding species difference elsewhere (page 9), this has not been corrected on page 12 (L535).

New comments (line numbers refer to the new version):

  • I am enthusiastic about the use of a table (Table 1) to outline the cellular effects of GLP-1. However, this table would benefit from some changes:
    • I suggest the authors include references to the literature.
    • It would be helpful to have concentrations used in the stimulation protocol indicated in the figure as well.
    • The table only covers responses to GLP1R stimulation in hypothalamic neurons. Is there a particular reason for this? Do the authors not think including other areas would be informative? If the authors choose to only include hypothalamic neurons, it would be beneficial to indicate the region of the hypothalamus (e.g. PVN, LH, Arc, DMH).
  • L66: Add reference to mapping of GLP-1 neurons in humans (Zheng et al., 2015).
  • L73: ‘However, GLP-1R nerve fibers are observed throughout the hypothalamus in rodents with a notable difference in the arcuate nucleus versus primates.’ What is meant by ‘GLP-1R nerve fibers’?
  • L83: The extended amygdala includes the BNST and CeA, but the authors seem to suggest it is a structure independent of the CeA and BNST. Please rephrase.
  • ‘GLP-1R mRNA is expressed in pancreatic islets (in β, α and δ-cells predominantly).’ Beta, alpha, and delta cells are the cells of the pancreatic islets. It is true that GLP1R is expressed in all, although at very low levels in alpha cells. This sentence could be more precise. See (Ast et al., 2020) for exact (experimental) numbers.
  • L119: References 21, 64, 67, 68. Please check the appropriateness of their placement here. It appears that only (Ast et al., 2020) should be referenced here and the others belong in the previous sentence.
  • L127: ‘The GLP-1 receptor is primarily expressed in neurons.’ Please provide a reference for this statement. GLP-1R has been demonstrated in astrocytes by several labs.
  • L204: ‘All members of the GLP-1R family increase cAMP levels’. There is only one GLP-1R. What do the authors mean here?
  • L205: ‘[…] glucocorticoids also increase the level of cAMP in both mice and humans’. Where are the levels of cAMP increased? Glucocorticoids act throughout the body. Where was this measured?
  • L340: ‘However, the central nucleus of the amygdala (CeA), a brain region essential for the initiation of the stress response [161], appears critical for generating the anxiogenic effects of GLP-1 since the administration of the peptide in CeA does not modify plasma corticosterone levels but increases the time spent in the open arms of the elevated plus maze (EMP).’ This is incorrect. GLP-1 decreases the time spent in the open arms, while the antagonist, Exendin-9, increases the time spent in the open arms.
  • L362: ‘[…] PPG downregulation and long-lasting PVN-GLP-1 fiber staining.’ What do the authors mean here? Is GLP-1 labelling in the PVN up- or downregulated following chronic stress?
  • L380: ‘[…] GLP-1 positive neurons, including those of the PVN in rats.’ There are no GLP-1 neurons in the PVN. Please rephrase.
  • L386: ‘[…] but LiCl does not block aversive effects […]’. LiCl induces aversion, it doesn’t block aversion. Please rephrase.
  • L389: ‘Furthermore, intra-amygdala administration GLP-1(9–36) prevents taste aversion learning in response to ip. injections of the LiCl’. This should be des-His1,Glu9-exendin-4, the GLP1R antagonist, not GLP-1(9-36), which is inactive.
  • L433: ‘[…] giving a possible explanation to the observation that exogenously administered GLP-1 only inhibits food intake after feeding’. Can you measure inhibition in food intake in animals that haven’t been fed? This statement needs to be more precise.
  • L435: The term ‘vagal neurons’ is ambiguous. It can refer to both motor and sensory neurons. Please specify.
  • L459: ‘However, GLP-1R activation in the CNS presents species-specific differences.’ Please give examples and references.
  • L462: ‘Recently it has been described that GLP1NTS cells represent a subset of LepRbNTS cells in mice, while in NTS rats, GLP1- and LepRb-containing cells are distinct [208].’
    • Please include a reference demonstrating that rat GLP1 neurons do not express leptin receptors.
    • While it is true that ref 208 is a recent paper, there have been several previous publications showing that leptin activates GLP1 neurons. Please include those.
    • Please be consistent with the names for cell types. Here the authors use GLP1NTS for the first time.
  • L469: Please check the appropriateness of the references provided. They do not appear to support the statement regarding inflammatory and stressful signals.
  • L502: ‘At physiologically relevant concentrations (pico- or nano-moL/L), both GLP-1 and Ex4 […]’. How do we know whether concentrations are physiologically relevant when applied ex vivo without having measured their levels in vivo? Please rephrase.
  • L505: Reference missing.
  • L505: How do the authors explain that inhibitory transmission in the PVN leads to decreased food intake and increased HPA axis activity?
  • L509: ‘Interestingly DPP4 inhibitors, which enhance the endogenous GLP-1 levels, also enhances GABAergic transmission.’ How were these inhibitors administered? In vivo? Ex vivo?
  • L528: ‘On the one hand, in enteric neurons, GABA exerts a depolarizing action and, therefore, a cellular activation.’ Reference missing.
  • L531: Has depolarisation in response to GABA in adult neurons been demonstrated? If so, please provide a reference.
  • L579: Reference missing.
  • L581: Reference missing.
  • L618: The conventional abbreviation for calcitonin gene-related peptide is CGRP.
  • L631: ‘Reward-related regions such as ventral tegmental areas and the NAc, the core as well the shell subregion receive NTS GLP-1 neuronal projections.’ (Merchenthaler et al., 1999) is not appropriate here. Instead include (Llewellyn-Smith et al., 2011) as well as the already cited publications.
  • L650: ‘GLP-1 R stimulation reduces food palatability’. It reduced the reward associated, not the palatability itself. The palatability is a feature of the diet.

Ast, J., Arvaniti, A., Fine, N. H. F., Nasteska, D., Ashford, F. B., Stamataki, Z., Koszegi, Z., Bacon, A., Jones, B. J., Lucey, M. A., Sasaki, S., Brierley, D. I., Hastoy, B., Tomas, A., D’Agostino, G., Reimann, F., Lynn, F. C., Reissaus, C. A., Linnemann, A. K., … Hodson, D. J. (2020). Super-resolution microscopy compatible fluorescent probes reveal endogenous glucagon-like peptide-1 receptor distribution and dynamics. Nature Communications, 11(1), 467. https://doi.org/10.1038/s41467-020-14309-w

Holt, M. K., Pomeranz, L. E., Beier, K. T., Reimann, F., Gribble, F. M., & Rinaman, L. (2019). Synaptic inputs to the mouse dorsal vagal complex and its resident preproglucagon neurons. The Journal of Neuroscience: The Official Journal of the Society for Neuroscience. https://doi.org/10.1523/JNEUROSCI.2145-19.2019

Kanoski, S. E., Rupprecht, L. E., Fortin, S. M., De Jonghe, B. C., & Hayes, M. R. (2012). The role of nausea in food intake and body weight suppression by peripheral GLP-1 receptor agonists, exendin-4 and liraglutide. Neuropharmacology, 62(5–6), 1916–1927. https://doi.org/10.1016/j.neuropharm.2011.12.022

Lachey, J. L., D’Alessio, D. A., Rinaman, L., Elmquist, J. K., Drucker, D. J., & Seeley, R. J. (2005). The Role of Central Glucagon-Like Peptide-1 in Mediating the Effects of Visceral Illness: Differential Effects in Rats and Mice. Endocrinology, 146(1), 458–462. https://doi.org/10.1210/en.2004-0419

Llewellyn-Smith, I. J., Gnanamanickam, G. J., Reimann, F., Gribble, F. M., & Trapp, S. (2013). Preproglucagon (PPG) neurons innervate neurochemically identified autonomic neurons in the mouse brainstem. Neuroscience, 229, 130–143. https://doi.org/10.1016/j.neuroscience.2012.09.071; 10.1016/j.neuroscience.2012.09.071

Llewellyn-Smith, I. J., Marina, N., Manton, R. N., Reimann, F., Gribble, F. M., & Trapp, S. (2015). Spinally projecting preproglucagon axons preferentially innervate sympathetic preganglionic neurons. Neuroscience, 284, 872–887. https://doi.org/10.1016/j.neuroscience.2014.10.043

Llewellyn-Smith, I. J., Reimann, F., Gribble, F. M., & Trapp, S. (2011). Preproglucagon neurons project widely to autonomic control areas in the mouse brain. Neuroscience, 180, 111–121. https://doi.org/10.1016/j.neuroscience.2011.02.023; 10.1016/j.neuroscience.2011.02.023

Merchenthaler, I., Lane, M., & Shughrue, P. (1999). Distribution of pre-pro-glucagon and glucagon-like peptide-1 receptor messenger RNAs in the rat central nervous system. J Comp Neurol, 403, 261–280.

Peng, J., Long, B., Yuan, J., Peng, X., Ni, H., Li, X., Gong, H., Luo, Q., & Li, A. (2017). A Quantitative Analysis of the Distribution of CRH Neurons in Whole Mouse Brain. Frontiers in Neuroanatomy, 11. https://doi.org/10.3389/fnana.2017.00063

Zheng, H., Cai, L., & Rinaman, L. (2015). Distribution of glucagon-like peptide 1-immunopositive neurons in human caudal medulla. Brain Struct Funct, 220(2), 1213–1219. https://doi.org/10.1007/s00429-014-0714-z

Author Response

We thank the reviewer for his/her positive comments. We greatly appreciate all comments of the referee that forced us to extensively reconstruct the manuscript with a significant improvement in this final form.

Broad comments:

  • The flow and logic of the manuscript is still somewhat confusing. Future readers would benefit from a more logical discussion of what the individual paragraphs tell us about the topic of the review: GLP-1 in stress.
  • Fig 1 remains unclear and inaccurate to this reader. While the authors added an arrow indicating vagal input to the NTS, the figure still appears to suggest that vagal afferents transmit GLP1 peripheral GLP-1 signalling to the NTS via the AP. Can the authors provide references to the literature this is based on?

We agree with the very pertinent comment of the referee. The figure and text in the result section were accordingly changed.

  • In addition, the authors state in the figure legend that the schematic shows effects of GLP-1 peripherally. That is mostly not the case (bar the effect on sensory vagal input), so I suggest the authors rephrase the description of the figure.

We have rephrased the description of Fig 1.

  • Abbreviations are used inconsistently. As an example, SNS is used to denote 'sympathetic nervous system', yet the full term 'sympathetic nervous system' appears eight times in the text. Please check for this type of error throughout the manuscript.

Thanks, we have checked the abbreviations.

  • The revised version should be carefully edited to remove grammatical/typographical errors, missing or duplicated words, etc.

Following the referee's suggestion, we edited the manuscript to correct possible grammatical/typographical errors, missing or duplicates.

  • I found at least one example of a duplicated reference (133 and 135). Please double check the bibliography.

We have checked the bibliography and eliminated the duplicates

Specific comments that were incompletely addressed in the revision (line numbers refer to the previous version):

  • P1L38: I do not like the term 'the homeostatic system' in this place. The homeostatic system cannot act on the hypothalamic and brainstem circuits. If anything, those circuits are part of'the homeostatic system'. Homeostasis is a state of equilibrium, not a system like the sympathetic nervous system or the central GLP-1 system.

'The homeostatic mechanism' is also not a correct term. I would completely rephrase the sentence to avoid confusion.

We have changed the term, and we have rephrased the sentence.

  • P2L59: 'The neurons of the NTS mainly project efferents to the PVN in the hypothalamus [14], providing glutamatergic input.' It is correct that the NTS provides massive input to the PVN, catecholaminergic as well as glutamatergic! However, the projections are from the NTS are widespread! E.g. the PBN is a dense target too. In addition, it would seem more relevant to discuss the widespread projection patterns of GCG neurons here (Llewellyn-Smith et al., 2011, 2013, 2015).

This comment was not addressed.

The paragraph has been completely modified in the second version and attending to the referee's suggestion, we have rewritten this part of the manuscript in this new version.

  • P6L243: This paragraph is very confusing. Are the authors saying that GABA induces release of GLP-1 from L cells? And that beta-cells release GABA in response to GLP-1? Please rephrase and explain the relevance to stress.

This is still very confusing and has not been addressed. I fail to see how the two statements are related. One is talking about the effect of GABA on GLP-1-producing cells. The next about the effect of GLP-1 on GABA release. And all of this in the periphery. Can the authors make these statements clearer and indicate their relevance to the topic of the manuscript?

We agree with the referee. To focus on the topic, we have removed the paragraph related to the peripheral effects of GABA. We were keeping only the hypothalamic effects of it.

  • P9L364: Compared to other hormones (e.g. CCK) a day is actually quite a long-lasting effect.

This is still incorrectly described in the text (L455). Unless large doses of CCK are used (and these induce visceral illness) then the effect of CCK on food intake is very brief (only clear at 30mins post-injection). The effect of liraglutide is comparatively long.

As the referee has suggested, we have modified this sentence.

  • P11L430: How do the authors explain this discrepancy between liraglutide and GLP-1?

The authors did not address my concern. Now L592.

We have addressed this comment of the referee in the new version of the manuscript.

  • P12L512: CRF is actually expressed in Barrington's nucleus adjacent to the locus coeruleus. Only very little is present in LC. See e.g. (Holt et al., 2019; Peng et al., 2017).

This was not corrected in the text, which still states that CRF is expressed in locus coeruleus. Please change.  

Following the referee's suggestion, It has been adequately corrected in the text.

  • P12L526-527: This is only true for rats. (Lachey et al., 2005).

While the authors do include the information regarding species difference elsewhere (page 9), this has not been corrected on page 12 (L535).

We have included this information in the text.

New comments (line numbers refer to the new version):

  • I am enthusiastic about the use of a table (Table 1) to outline the cellular effects of GLP-1. However, this table would benefit from some changes:
    • I suggest the authors include references to the literature.
    • It would be helpful to have concentrations used in the stimulation protocol indicated in the figure as well.
    • The table only covers responses to GLP1R stimulation in hypothalamic neurons. Is there a particular reason for this? Do the authors not think including other areas would be informative? If the authors choose to only include hypothalamic neurons, it would be beneficial to indicate the region of the hypothalamus (e.g. PVN, LH, Arc, DMH).

According to referee's suggestion, we have added additional information to the Table.

  • L66: Add reference to mapping of GLP-1 neurons in humans (Zheng et al., 2015).

The reference has been adequately added.

  • L73: 'However, GLP-1R nerve fibers are observed throughout the hypothalamus in rodents with a notable difference in the arcuate nucleus versus primates.' What is meant by 'GLP-1R nerve fibers'?

We thank the reviewer for the comment, we have noticed the mistake, and we have correct it in the text.

  • L83: The extended amygdala includes the BNST and CeA, but the authors seem to suggest it is a structure independent of the CeA and BNST. Please rephrase.

We have rephrased the sentence.

  • 'GLP-1R mRNA is expressed in pancreatic islets (in β, α, and δ-cells predominantly).' Beta, alpha, and delta cells are the cells of the pancreatic islets. It is true that GLP1R is expressed in all, although at very low levels in alpha cells. This sentence could be more precise. See (Ast et al., 2020) for exact (experimental) numbers.

We have added the exact experimental numbers of mRNA expression percentage from the paper of Ast and colleagues.

  • L119: References 21, 64, 67, 68. Please check the appropriateness of their placement here. It appears that only (Ast et al., 2020) should be referenced here and the others belong in the previous sentence.

We have replaced the reference to the correct place in the paragraph.

  • L127: 'The GLP-1 receptor is primarily expressed in neurons.' Please provide a reference for this statement. GLP-1R has been demonstrated in astrocytes by several labs.

We have added the references, as the referee has suggested.

  • L204: 'All members of the GLP-1R family increase cAMP levels'. There is only one GLP-1R. What do the authors mean here?

We thank the referee for the notice. We have corrected the mistake.

  • L205: '[…] glucocorticoids also increase the level of cAMP in both mice and humans'. Where are the levels of cAMP increased? Glucocorticoids act throughout the body. Where was this measured?

Lines 204 and 205: We have changed the paragraph, in the hope that it is better understood.

  • L340: 'However, the central nucleus of the amygdala (CeA), a brain region essential for the initiation of the stress response [161], appears critical for generating the anxiogenic effects of GLP-1 since the administration of the peptide in CeA does not modify plasma corticosterone levels but increases the time spent in the open arms of the elevated plus maze (EMP).' This is incorrect. GLP-1 decreases the time spent in the open arms, while the antagonist, Exendin-9, increases the time spent in the open arms.

We have noticed the mistake, and we have correct it in the text.

  • L362: '[…] PPG downregulation and long-lasting PVN-GLP-1 fiber staining.' What do the authors mean here? Is GLP-1 labelling in the PVN up- or downregulated following chronic stress?

We have correct this mistake in the text. We thanks the referee for the comment.

  • L380: '[…] GLP-1 positive neurons, including those of the PVN in rats.' There are no GLP-1 neurons in the PVN. Please rephrase.

We have rephrased the sentence.

  • L386: '[…] but LiCl does not block aversive effects […]'. LiCl induces aversion, it doesn't block aversion. Please rephrase.

We have rephrased the sentence.

  • L389: 'Furthermore, intra-amygdala administration GLP-1(9–36) prevents taste aversion learning in response to ip. injections of the LiCl'. This should be des-His1,Glu9-exendin-4, the GLP1R antagonist, not GLP-1(9-36), which is inactive.

We have noticed the mistake, and we have correct it in the text.

  • L433: '[…] giving a possible explanation to the observation that exogenously administered GLP-1 only inhibits food intake after feeding'. Can you measure inhibition in food intake in animals that haven't been fed? This statement needs to be more precise.

We have added clarifications to the text to be more precise.

  • L435: The term 'vagal neurons' is ambiguous. It can refer to both motor and sensory neurons. Please specify.

We have specified the correct term in the text.

  • L459: 'However, GLP-1R activation in the CNS presents species-specific differences.' Please give examples and references.

We have given more examples, and we have added the correspondent references.

  • L462: 'Recently it has been described that GLP1NTScells represent a subset of LepRbNTS cells in mice, while in NTS rats, GLP1- and LepRb-containing cells are distinct [208].'

Please include a reference demonstrating that rat GLP1 neurons do not express leptin receptors.

While it is true that ref 208 is a recent paper, there have been several previous publications showing that leptin activates GLP1 neurons. Please include those.

We have modified the paragraph in order to respond to all these concerns.

  • Please be consistent with the names for cell types. Here the authors use GLP1NTSfor the first time.

We have unified the denomination of the cell types.

  • L469: Please check the appropriateness of the references provided. They do not appear to support the statement regarding inflammatory and stressful signals.

We thank the referee for the notice. We have added the references regarding inflammatory and stressful signals.

  • L502: 'At physiologically relevant concentrations (pico- or nano-moL/L), both GLP-1 and Ex4 […]'. How do we know whether concentrations are physiologically relevant when applied ex vivo without having measured their levels in vivo? Please rephrase.

We have rephrased the paragraph.

  • L505: Reference missing.

We have added the reference.

  • L505: How do the authors explain that inhibitory transmission in the PVN leads to decreased food intake and increased HPA axis activity?

We have verified that it was not correctly written, and we have modified the paragraph to clear it.

  • L509: 'Interestingly DPP4 inhibitors, which enhance the endogenous GLP-1 levels, also enhances GABAergic transmission.' How were these inhibitors administered? In vivoEx vivo?

Added in vivo.

  • L528: 'On the one hand, in enteric neurons, GABA exerts a depolarizing action and, therefore, a cellular activation.' Reference missing.

The paragraph has been removed.

  • L531: Has depolarisation in response to GABA in adult neurons been demonstrated? If so,

please provide a reference.

The paragraph has been removed.

  • L579: Reference missing.

We have added the reference.

L581: Reference missing.

We have added the reference.

  • L618: The conventional abbreviation for calcitonin gene-related peptide is CGRP.

We have noticed the mistake, and we have correct it in the text.

  • L631: 'Reward-related regions such as ventral tegmental areas and the NAc, the core as well the shell subregion receive NTS GLP-1 neuronal projections.' (Merchenthaler et al., 1999) is not appropriate here. Instead include (Llewellyn-Smith et al., 2011) as well as the already cited publications.

We have included the appropriate reference.

  • L650: 'GLP-1 R stimulation reduces food palatability'. It reduced the reward associated, not the palatability itself. The palatability is a feature of the diet.

We thank the referee for the notice. We have corrected the mistake.

Ast, J., Arvaniti, A., Fine, N. H. F., Nasteska, D., Ashford, F. B., Stamataki, Z., Koszegi, Z., Bacon, A., Jones, B. J., Lucey, M. A., Sasaki, S., Brierley, D. I., Hastoy, B., Tomas, A., D'Agostino, G., Reimann, F., Lynn, F. C., Reissaus, C. A., Linnemann, A. K., … Hodson, D. J. (2020). Super-resolution microscopy compatible fluorescent probes reveal endogenous glucagon-like peptide-1 receptor distribution and dynamics. Nature Communications11(1), 467. https://doi.org/10.1038/s41467-020-14309-w

Holt, M. K., Pomeranz, L. E., Beier, K. T., Reimann, F., Gribble, F. M., & Rinaman, L. (2019). Synaptic inputs to the mouse dorsal vagal complex and its resident preproglucagon neurons. The Journal of Neuroscience: The Official Journal of the Society for Neuroscience. https://doi.org/10.1523/JNEUROSCI.2145-19.2019

Kanoski, S. E., Rupprecht, L. E., Fortin, S. M., De Jonghe, B. C., & Hayes, M. R. (2012). The role of nausea in food intake and body weight suppression by peripheral GLP-1 receptor agonists, exendin-4 and liraglutide. Neuropharmacology62(5–6), 1916–1927. https://doi.org/10.1016/j.neuropharm.2011.12.022

Lachey, J. L., D'Alessio, D. A., Rinaman, L., Elmquist, J. K., Drucker, D. J., & Seeley, R. J. (2005). The Role of Central Glucagon-Like Peptide-1 in Mediating the Effects of Visceral Illness: Differential Effects in Rats and Mice. Endocrinology146(1), 458–462. https://doi.org/10.1210/en.2004-0419

Llewellyn-Smith, I. J., Gnanamanickam, G. J., Reimann, F., Gribble, F. M., & Trapp, S. (2013). Preproglucagon (PPG) neurons innervate neurochemically identified autonomic neurons in the mouse brainstem. Neuroscience229, 130–143. https://doi.org/10.1016/j.neuroscience.2012.09.071; 10.1016/j.neuroscience.2012.09.071

Llewellyn-Smith, I. J., Marina, N., Manton, R. N., Reimann, F., Gribble, F. M., & Trapp, S. (2015). Spinally projecting preproglucagon axons preferentially innervate sympathetic preganglionic neurons. Neuroscience284, 872–887. https://doi.org/10.1016/j.neuroscience.2014.10.043

Llewellyn-Smith, I. J., Reimann, F., Gribble, F. M., & Trapp, S. (2011). Preproglucagon neurons project widely to autonomic control areas in the mouse brain. Neuroscience180, 111–121. https://doi.org/10.1016/j.neuroscience.2011.02.023; 10.1016/j.neuroscience.2011.02.023

Merchenthaler, I., Lane, M., & Shughrue, P. (1999). Distribution of pre-pro-glucagon and glucagon-like peptide-1 receptor messenger RNAs in the rat central nervous system. J Comp Neurol403, 261–280.

Peng, J., Long, B., Yuan, J., Peng, X., Ni, H., Li, X., Gong, H., Luo, Q., & Li, A. (2017). A Quantitative Analysis of the Distribution of CRH Neurons in Whole Mouse Brain. Frontiers in Neuroanatomy11. https://doi.org/10.3389/fnana.2017.00063

Zheng, H., Cai, L., & Rinaman, L. (2015). Distribution of glucagon-like peptide 1-immunopositive neurons in human caudal medulla. Brain Struct Funct220(2), 1213–1219. https://doi.org/10.1007/s00429-014-0714-z

Reviewer 3 Report

I feel the authors did an excellent job conveying the known information regarding CNS derived GLP-1 as a neuromodulator and how this signaling may be dysregulated in obese and T2D rodents and humans. I really do not have much feedback for the authors other than adjusting Figure 1 by including the acronym definitions along with the Figure 1 legend.

Author Response

We would like to thank the reviewer for the positive comments about our manuscript. We have adjusted figure 1 as he/she has suggested.

This manuscript is a resubmission of an earlier submission. The following is a list of the peer review reports and author responses from that submission.

Round 1

Reviewer 1 Report

Glucagon-like peptide-1 (GLP-1) is an incretin and neuropeptide that plays a major role in widespread physiological processes, including, but not limited to, autonomic, neuroendocrine, and behavioural responses to stress (S. Ghosal et al., 2013; Sriparna Ghosal et al., 2017; M. K. Holt & Trapp, 2016; Kinzig et al., 2003; Maniscalco et al., 2015; Maniscalco & Rinaman, 2017; Terrill et al., 2018, 2019; Diana L. Williams et al., 2018; Zhang et al., 2010, 2010; Zheng et al., 2019). GLP-1 is produced by cells in both the gut and the brain and recent findings (referenced above) have placed the central source of GLP-1, the GCG neurons, at the centre of responses to stress. The aim of this review is to provide an up-to-date summary of the role of GLP-1 in HPA axis regulation, food intake control, stress responses, and modulation of sympathetic activity.

I found the topic timely and well-suited to the readership of the journal. However, I found the manuscript disorganised and hard to follow. There was little clear direction or intention and I did not find the title or abstract representative of the main text. There was scarce relevant discussion of the role of central GLP-1 in stress and how this might relate to the positive relationship between chronic stress and obesity. Furthermore, there was a lack of review of the most recent literature on the subject and important, relevant references were frequently missing. I did not find the figures helpful nor clear. I hope the authors find my comments below constructive.

Broad comments:

  1. The title could be improved: ‘Glucagon-like Peptide-1 (GLP-1) in the integration of neural and endocrine responses to feeding under stress’. What is meant by ‘responses to feeding under stress’? The secretion/release of GLP-1 in response to feeding is barely discussed (and indeed is not straight-forward in the brain(Kreisler et al., 2014; Kreisler & Rinaman, 2016)). Do the authors mean to say ‘Glucagon-like Peptide-1 (GLP-1) in the integration of neural and endocrine responses to stress’?
  2. The topic of this review is the role of GLP-1 in stress regulation. However, stress is only sporadically mentioned and indeed much of the review focusses on the intracellular signalling mechanisms downstream of the GLP-1 receptor. While this is certainly an interesting and important topic, I was not convinced of its relevance to the topic of the review as stated by the authors. I suggest the authors reconsider the in-depth summary of GLP-1 receptor signalling in other cell types than neurons unless they can justify the relevance to stress. A brief description is appropriate to highlight the potential differences in signalling between brain and pancreas, but as a reader I was left confused by the apparent lack of relevance for stress modulation. Instead a thorough review of the distribution of GLP-1 receptor expression in the brain (see (Cork et al., 2015; Graham et al., 2020; Heppner et al., 2015; Merchenthaler et al., 1999)) and what is currently known about the signalling mechanisms and cellular effects of GLP-1 in different brain regions would be very interesting.
  3. Although the declared topic of the review is the role of GLP-1 in stress, few words are devoted to discussing the role of the endogenous source of GLP-1 within the brain, the GCG neurons. The authors briefly mention these neurons but miss the opportunity to discuss their role in stress regulation. While the literature on this topic is still in its early stages, there are several independent research articles highlighting the importance of GCG neurons in stress regulation. A discussion of GCG neurons, their projection patterns (Llewellyn-Smith et al., 2011, 2013, 2015; Niels Vrang et al., 2007), their regulation in response to many stress-related signals (Anesten et al., 2016; Maniscalco et al., 2015; Rinaman, 1999a; Terrill et al., 2019; N. Vrang et al., 2003), their central inputs, which include areas activated in response to stress (Marie K. Holt, Pomeranz, et al., 2019), their ability to suppress feeding (Gaykema et al., 2017; Marie K. Holt, Richards, et al., 2019; Liu et al., 2017), and their role in food intake suppression in response to stress (Marie K. Holt, Richards, et al., 2019; Terrill et al., 2019; Diana L. Williams et al., 2018).
  4. There is a growing body of literature on the effect of GLP-1 or GLP-1 receptor agonists on sympathetic activity (Baggio et al., 2017; Sriparna Ghosal et al., 2017; Marie K. Holt et al., 2020; Yamamoto et al., 2002, 2003). The authors briefly mention increases in heart rate and blood pressure in response to GLP-1 receptor activation, but miss the opportunity to link this to stress, when they don’t discuss papers finding that GLP-1 receptors in the PVN are necessary for stress-induced tachycardia (Sriparna Ghosal et al., 2017), that sympathetic activity is necessary in tachycardic responses to Exendin-4 (Baggio et al., 2017; Marie K. Holt et al., 2020), and that direct application of GLP-1 to the spinal cord increases heart rate (Marie K. Holt et al., 2020), suggesting direct activation of the sympathetic nervous system.
  5. Similar to the above, there is significant evidence for a role of GLP-1 in anxiety-like behaviour, but this is only sporadically mentioned. When discussing the role of GLP-1 in stress it seems prudent to thoroughly review the effects of GLP-1 on anxiety-like behaviours, which appear to be mediated by many different brain regions (Kinzig et al., 2003; López-Ferreras et al., 2020; Terrill et al., 2018, 2019; Zheng et al., 2019).
  6. I did not find the figures very clear. Figure 1 in particular was unclear to me. Why do the vagal afferents appear to terminate in the area postrema (AP), when the bulk of the input goes to the NTS? In addition, the diagram suggests to me that the authors think that GLP-1 released from the gut activates vagal afferents, which in turn activates GLP-1 receptor-expressing neurons in the AP. To my knowledge there is no evidence for this or for the suggestion that the AP transmits that signal to the NTS. In fact, a recent paper found that the AP is not necessary for activation of the NTS following peripheral administration of a GLP-1 receptor analogue (Fortin et al., 2020).
  7. The organisation of the manuscript was difficult to follow throughout. I suggest the authors rearrange sections to make it clearer to the reader how each discussion of GLP-1 function relates to the topic of the review: GLP-1 in stress. It is often unclear if the authors are focussing on peripheral or central GLP-1 and whether they consider these systems to be working independently or in concert. Similarly, it was not obvious which species the authors were referring when discussing findings. Considering the species differences observed (see e.g. (Huo et al., 2008; Lachey et al., 2005; Rinaman, 1999b)) and the readership of the journal, which includes many researchers focussing on human studies, it is particularly important to be diligent with species information.
  8. There were several abbreviations that did not appear to be defined previously, or the full term was used after abbreviations were defined.

Specific comments:

P1L38: I do not like the term ‘the homeostatic system’ in this place. The homeostatic system cannot act on the hypothalamic and brainstem circuits. If anything, those circuits are part of ‘the homeostatic system’. Homeostasis is a state of equilibrium, not a system like the sympathetic nervous system or the central GLP-1 system.

P1L42: This sentence appears incomplete to me. Please rephrase.

P2L51: ‘The glucagon-like peptide-1…’ Delete ‘The’.

P2L56-59: ‘The primary source of endogenous GLP-1 within the brain is a population of non-catecholaminergic neurons in the caudal portion of NTS [13]. Also, neurons in the caudal brainstem and some neurons in the adjacent dorsomedial part of the medullary reticular nucleus can produce GLP-1.’ 2 points:

  • Several more references should be included here: (Hisadome et al., 2010; Larsen et al., 1997; Merchenthaler et al., 1999).
  • The second sentence does not make sense to me. It suggests (with ‘also’) that these areas are in addition to the NTS, however the NTS is in the caudal brainstem. Please rephrase.

P2L59: ‘The neurons of the NTS mainly project efferents to the PVN in the hypothalamus [14], providing glutamatergic input.’ It is correct that the NTS provides massive input to the PVN, catecholaminergic as well as glutamatergic! However, the projections are from the NTS are widespread! E.g. the PBN is a dense target too. In addition, it would seem more relevant to discuss the widespread projection patterns of GCG neurons here (Llewellyn-Smith et al., 2011, 2013, 2015).

P3L77: ‘oral glucose intake for intravenous glucose infusion’. ‘for’ should be ‘or’.

P3L98: It would be better to include a reference to the actual papers rather than the (otherwise fantastic) review by Jens Holst.

P3L102: And mice! Indeed reference 43 was a study in mice (Ast et al., 2020) and in fact did not aim to map the distribution of GLP-1 receptors, but to identify central populations accessible to a peripherally administered GLP-1 receptor antagonist. It would be appropriate here to include citations of papers mapping the distribution of GLP-1 receptors in mice (Cork et al., 2015; Graham et al., 2020).

P3L110: It would be good to mention the limitation of GLP-1 receptor antibodies here.

P4L130: Do the authors not think that the main reason GLP-1 has different effects in different tissues is the very different cell types activated there. Would you ever expect a peptide to have similar final outputs when acting in brain and pancreas? Perhaps it occurs, but surely would be unexpected. More interesting would be a discussion of why GLP-1 appears to have inhibitory effects on a cellular level in some neuronal subtypes, but excitatory in others.  

P4L141: ‘Most of the already identify actions of GLP-1 analogs in the nervous system are vinculated to changes in the activation of different groups of neurons, most of them located in areas of the hypothalamus.’ A lot of effects of GLP-1 and analogues have been identified outside the hypothalamus! Indeed, sites responsive to GLP-1 are widespread! Please rephrase.

P4L150: I suggest the authors include a table outlining the different cellular effects of GLP-1 or analogues in different brain regions. That would provide a clear overview for the reader.  

P5L195: While it is true that a few studies have found antidepressant effects of GLP-1 receptor stimulation, many studies have found that GLP-1 is anxiogenic (Kinzig et al., 2003; López-Ferreras et al., 2020; Maniscalco et al., 2015; Zheng et al., 2019). For completeness, this should be acknowledged here.

P6L240: Has the paraventricular nucleus already been defined elsewhere (PVN)? Is this the hypothalamic or thalamic paraventricular nucleus?

P6L243: This paragraph is very confusing. Are the authors saying that GABA induces release of GLP-1 from L cells? And that beta-cells release GABA in response to GLP-1? Please rephrase and explain the relevance to stress.

P7L259: Please also include work by Liu et al. showing clear involvement of AMPA/Kainate receptors in response to GLP-1 in the PVN (Liu et al., 2017).

P7L281: Please include references demonstrating input from A2 and GLP-1 neurons to CRF-neurons in the hypothalamus, respectively.

P7L282: ‘revealing the role of NTS to regulate the stress responses’. This does not really follow. The PVN has many other functions too. Please rephrase.

P7L283: I am not sure what is meant by ‘target-specific’ in this sentence? If anything, GLP-1 neurons appear to collateralize substantially.

P7L287: The cited article does not support this claim.

P7L291: Please include a mention of the excellent and highly relevant work by Liu et al. already cited elsewhere (Liu et al., 2017).

P7L298: ‘Moreover, rats treated with Ex4 spent more time in the open arms of the elevated plus-maze (EPM), indicating that the central GLP-1 system mediates the anxiety-like behavior [93] (Figure 1).’ This should be less time in the open arms!

P8L308: Astressin is not selective for CRHR1. Please rephrase.

P8L320: Please also include discussion of this paper showing a role for endogenous GLP-1 in the BST in stress-induced hypophagia in mice (Diana L. Williams et al., 2018).

P8329: There is a lack of discussion of studies investigating the role of the sympathetic nervous system in the cardiovascular effects of GLP-1 receptor stimulation. See e.g. (Baggio et al., 2017; Griffioen et al., 2011; Marie K. Holt et al., 2020; Yamamoto et al., 2003). In addition, studies looking at thermogenic effects of GLP-1 receptor stimulation would be relevant here. See e.g. (Beiroa et al., 2014; Lee et al., 2018; Lockie et al., 2012).

P8L332: GCG neurons also project directly to the spinal cord (Llewellyn-Smith et al., 2015).

P8L333: Reference missing.

P8L334: GLP-1 neurons are not catecholaminergic (Hisadome et al., 2010). Please rephrase. If the authors meant ‘GLP-1 receptor-expressing catecholaminergic neurons’ that would also be incorrect, since Yamamoto et al. (Yamamoto et al., 2003) did not demonstrate the expression of GLP-1 receptors on cFOS positive neurons and the activation of the AP could be indirect. While this would be surprising given the density of GLP-1 receptor-expressing neurons in the AP it nonetheless is a possibility that would have to be excluded.  

P8L340: Add (Llewellyn-Smith et al., 2015) to this discussion! It is highly relevant to this section that GCG neurons project directly to the spinal cord. In addition, direct application of GLP-1 to the spinal cord increases heart rate (Marie K. Holt et al., 2020).

P8L350: What is ‘SAS’?

P9L364: Compared to other hormones (e.g. CCK) a day is actually quite a long-lasting effect.

P9L369: What is meant by ‘them’?

P9L369: What is the relative contribution of peripheral and centrally derived GLP-1 to the control of food intake? There is a lot more to discuss about centrally derived GLP-1 and in particular the GCG neurons. These are silenced by fasting (Maniscalco et al., 2015) and drive suppression in feeding in response to stress (Marie K. Holt, Richards, et al., 2019). Why is that not mentioned here?

P9L385: These are conflicting statements. How do the authors explain this apparent discrepancy?

P9L390: GLP-1 neurons also project directly to areas linked to reward (BST, NAc, LS). (Llewellyn-Smith et al., 2011).

P10L425: Add relevant citations: (Cork et al., 2015; Graham et al., 2020; Llewellyn-Smith et al., 2011). The cited reference (Merchenthaler et al., 1999) does not demonstrate projections of GCG neurons.

P11L430: How do the authors explain this discrepancy between liraglutide and GLP-1?

P12L488: And anxiety-like behaviour! (López-Ferreras et al., 2020).

P12L497: I do not follow this sentence. Please rephrase.

P12L512: CRF is actually expressed in Barrington’s nucleus adjacent to the locus coeruleus. Only very little is present in LC. See e.g. (Marie K. Holt, Pomeranz, et al., 2019; Peng et al., 2017).

P12L526-527: This is only true for rats. (Lachey et al., 2005).

P13L539: And in the lateral septum in rats (Terrill et al., 2018) and mice (Terrill et al., 2019).

P13L555: This section is lacking discussion of the following points:  

  • Changes in PPG mRNA in a binge eating model (Mukherjee et al., 2020)
  • Changes in sensitivity to satiety hormones (e.g. Amylin) in a binge eating model (Maske et al., 2020)

P13L575: Please have a look at these very nice papers on the matter: (Mul et al., 2013; D. L. Williams et al., 2011).

References

Anesten, F., Holt, M. K., Schele, E., Palsdottir, V., Reimann, F., Gribble, F. M., Safari, C., Skibicka, K. P., Trapp, S., & Jansson, J. O. (2016). Preproglucagon neurons in the hindbrain have IL-6 receptor-alpha and show Ca2+ influx in response to IL-6. Am J Physiol Regul Integr Comp Physiol, 311(1), R115-23. https://doi.org/10.1152/ajpregu.00383.2015

Ast, J., Arvaniti, A., Fine, N. H. F., Nasteska, D., Ashford, F. B., Stamataki, Z., Koszegi, Z., Bacon, A., Jones, B. J., Lucey, M. A., Sasaki, S., Brierley, D. I., Hastoy, B., Tomas, A., D’Agostino, G., Reimann, F., Lynn, F. C., Reissaus, C. A., Linnemann, A. K., … Hodson, D. J. (2020). Super-resolution microscopy compatible fluorescent probes reveal endogenous glucagon-like peptide-1 receptor distribution and dynamics. Nature Communications, 11(1), 467. https://doi.org/10.1038/s41467-020-14309-w

Baggio, L. L., Ussher, J. R., McLean, B. A., Cao, X., Kabir, M. G., Mulvihill, E. E., Mighiu, A. S., Zhang, H., Ludwig, A., Seeley, R. J., Heximer, S. P., & Drucker, D. J. (2017). The autonomic nervous system and cardiac GLP-1 receptors control heart rate in mice. Molecular Metabolism, 6(11), 1339–1349. https://doi.org/10.1016/j.molmet.2017.08.010

Beiroa, D., Imbernon, M., Gallego, R., Senra, A., Herranz, D., Villarroya, F., Serrano, M., Fernø, J., Salvador, J., Escalada, J., Dieguez, C., Lopez, M., Frühbeck, G., & Nogueiras, R. (2014). GLP-1 agonism stimulates brown adipose tissue thermogenesis and browning through hypothalamic AMPK. Diabetes, 63(10), 3346–3358. https://doi.org/10.2337/db14-0302

Cork, S. C., Richards, J. E., Holt, M. K., Gribble, F. M., Reimann, F., & Trapp, S. (2015). Distribution and characterisation of Glucagon-like peptide-1 receptor expressing cells in the mouse brain. Molecular Metabolism. http://dx.doi.org/10.1016/j.molmet.2015.07.008

Fortin, S. M., Lipsky, R. K., Lhamo, R., Chen, J., Kim, E., Borner, T., Schmidt, H. D., & Hayes, M. R. (2020). GABA neurons in the nucleus tractus solitarius express GLP-1 receptors and mediate anorectic effects of liraglutide in rats. Science Translational Medicine, 12(533). https://doi.org/10.1126/scitranslmed.aay8071

Gaykema, R. P., Newmyer, B. A., Ottolini, M., Raje, V., Warthen, D. M., Lambeth, P. S., Niccum, M., Yao, T., Huang, Y., Schulman, I. G., Harris, T. E., Patel, M. K., Williams, K. W., & Scott, M. M. (2017). Activation of murine pre-proglucagon-producing neurons reduces food intake and body weight. The Journal of Clinical Investigation, 127(3), 1031–1045. https://doi.org/10.1172/JCI81335

Ghosal, S., Myers, B., & Herman, J. P. (2013). Role of central glucagon-like peptide-1 in stress regulation. Physiol Behav, 122, 201–207. https://doi.org/10.1016/j.physbeh.2013.04.003

Ghosal, Sriparna, Packard, A. E. B., Mahbod, P., McKlveen, J. M., Seeley, R. J., Myers, B., Ulrich-Lai, Y., Smith, E. P., D’Alessio, D. A., & Herman, J. P. (2017). Disruption of Glucagon-Like Peptide 1 Signaling in Sim1 Neurons Reduces Physiological and Behavioral Reactivity to Acute and Chronic Stress. Journal of Neuroscience, 37(1), 184–193. https://doi.org/10.1523/JNEUROSCI.1104-16.2016

Graham, D. L., Durai, H. H., Trammell, T. S., Noble, B. L., Mortlock, D. P., Galli, A., & Stanwood, G. D. (2020). A novel mouse model of glucagon-like peptide-1 receptor expression: A look at the brain. The Journal of Comparative Neurology. https://doi.org/10.1002/cne.24905

Griffioen, K. J., Wan, R., Okun, E., Wang, X., Lovett-Barr, M. R., Li, Y., Mughal, M. R., Mendelowitz, D., & Mattson, M. P. (2011). GLP-1 receptor stimulation depresses heart rate variability and inhibits neurotransmission to cardiac vagal neurons. Cardiovasc Res, 89(1), 72–78. https://doi.org/10.1093/cvr/cvq271

Heppner, K. M., Kirigiti, M., Secher, A., Paulsen, S. J., Buckingham, R., Pyke, C., Knudsen, L. B., Vrang, N., & Grove, K. L. (2015). Expression and Distribution of Glucagon-Like Peptide-1 Receptor mRNA, Protein and Binding in the Male Nonhuman Primate (Macaca mulatta) Brain. Endocrinology, 156(1), 255–267. https://doi.org/10.1210/en.2014-1675

Hisadome, K., Reimann, F., Gribble, F. M., & Trapp, S. (2010). Leptin directly depolarizes preproglucagon neurons in the nucleus tractus solitarius: Electrical properties of glucagon-like Peptide 1 neurons. Diabetes, 59, 1890–1898. https://doi.org/10.2337/db10-0128; 10.2337/db10-0128

Holt, M. K., & Trapp, S. (2016). The physiological role of the brain GLP-1 system in stress. Cogent Biol, 2(1). https://doi.org/10.1080/23312025.2016.1229086

Holt, Marie K., Cook, D. R., Brierley, D. I., Richards, J. E., Reimann, F., Gourine, A. V., Marina, N., & Trapp, S. (2020). PPG neurons in the nucleus of the solitary tract modulate heart rate but do not mediate GLP-1 receptor agonist-induced tachycardia in mice. Molecular Metabolism, 39, 101024. https://doi.org/10.1016/j.molmet.2020.101024

Holt, Marie K., Pomeranz, L. E., Beier, K. T., Reimann, F., Gribble, F. M., & Rinaman, L. (2019). Synaptic inputs to the mouse dorsal vagal complex and its resident preproglucagon neurons. The Journal of Neuroscience: The Official Journal of the Society for Neuroscience. https://doi.org/10.1523/JNEUROSCI.2145-19.2019

Holt, Marie K., Richards, J. E., Cook, D. R., Brierley, D. I., Williams, D. L., Reimann, F., Gribble, F. M., & Trapp, S. (2019). Preproglucagon Neurons in the Nucleus of the Solitary Tract Are the Main Source of Brain GLP-1, Mediate Stress-Induced Hypophagia, and Limit Unusually Large Intakes of Food. Diabetes, 68(1), 21–33. https://doi.org/10.2337/db18-0729

Huo, L., Gamber, K. M., Grill, H. J., & Bjørbaek, C. (2008). Divergent leptin signaling in proglucagon neurons of the nucleus of the solitary tract in mice and rats. Endocrinology, 149(2), 492–497. https://doi.org/10.1210/en.2007-0633

Kinzig, K. P., D’Alessio, D. A., Herman, J. P., Sakai, R. R., Vahl, T. P., Figueiredo, H. F., Murphy, E. K., & Seeley, R. J. (2003). CNS glucagon-like peptide-1 receptors mediate endocrine and anxiety responses to interoceptive and psychogenic stressors. J Neurosci, 23(15), 6163–6170.

Kreisler, A. D., Davis, E. A., & Rinaman, L. (2014). Differential activation of chemically identified neurons in the caudal nucleus of the solitary tract in non-entrained rats after intake of satiating vs. Non-satiating meals. Physiol Behav. https://doi.org/10.1016/j.physbeh.2014.01.015

Kreisler, A. D., & Rinaman, L. (2016). Hindbrain glucagon-like peptide-1 neurons track intake volume and contribute to injection stress-induced hypophagia in meal-entrained rats. Am J Physiol Regul Integr Comp Physiol, 310(10), R906-16. https://doi.org/10.1152/ajpregu.00243.2015

Lachey, J. L., D’Alessio, D. A., Rinaman, L., Elmquist, J. K., Drucker, D. J., & Seeley, R. J. (2005). The Role of Central Glucagon-Like Peptide-1 in Mediating the Effects of Visceral Illness: Differential Effects in Rats and Mice. Endocrinology, 146(1), 458–462. https://doi.org/10.1210/en.2004-0419

Larsen, P. J., Tang-Christensen, M., Holst, J. J., & Orskov, C. (1997). Distribution of glucagon-like peptide-1 and other preproglucagon-derived peptides in the rat hypothalamus and brainstem. Neuroscience, 77, 257–270.

Lee, S. J., Sanchez-Watts, G., Krieger, J.-P., Pignalosa, A., Norell, P. N., Cortella, A., Pettersen, K. G., Vrdoljak, D., Hayes, M. R., Kanoski, S. E., Langhans, W., & Watts, A. G. (2018). Loss of dorsomedial hypothalamic GLP-1 signaling reduces BAT thermogenesis and increases adiposity. Molecular Metabolism, 11, 33–46. https://doi.org/10.1016/j.molmet.2018.03.008

Liu, J., Conde, K., Zhang, P., Lilascharoen, V., Xu, Z., Lim, B. K., Seeley, R. J., Zhu, J. J., Scott, M. M., & Pang, Z. P. (2017). Enhanced AMPA Receptor Trafficking Mediates the Anorexigenic Effect of Endogenous Glucagon-like Peptide-1 in the Paraventricular Hypothalamus. Neuron, 96(4), 897-909.e5. https://doi.org/10.1016/j.neuron.2017.09.042

Llewellyn-Smith, I. J., Gnanamanickam, G. J., Reimann, F., Gribble, F. M., & Trapp, S. (2013). Preproglucagon (PPG) neurons innervate neurochemically identified autonomic neurons in the mouse brainstem. Neuroscience, 229, 130–143. https://doi.org/10.1016/j.neuroscience.2012.09.071; 10.1016/j.neuroscience.2012.09.071

Llewellyn-Smith, I. J., Marina, N., Manton, R. N., Reimann, F., Gribble, F. M., & Trapp, S. (2015). Spinally projecting preproglucagon axons preferentially innervate sympathetic preganglionic neurons. Neuroscience, 284, 872–887. https://doi.org/10.1016/j.neuroscience.2014.10.043

Llewellyn-Smith, I. J., Reimann, F., Gribble, F. M., & Trapp, S. (2011). Preproglucagon neurons project widely to autonomic control areas in the mouse brain. Neuroscience, 180, 111–121. https://doi.org/10.1016/j.neuroscience.2011.02.023; 10.1016/j.neuroscience.2011.02.023

Lockie, S. H., Heppner, K. M., Chaudhary, N., Chabenne, J. R., Morgan, D. A., Veyrat-Durebex, C., Ananthakrishnan, G., Rohner-Jeanrenaud, F., Drucker, D. J., DiMarchi, R., Rahmouni, K., Oldfield, B. J., Tschop, M. H., & Perez-Tilve, D. (2012). Direct control of brown adipose tissue thermogenesis by central nervous system glucagon-like peptide-1 receptor signaling. Diabetes, 61(11), 2753–2762. https://doi.org/10.2337/db11-1556

López-Ferreras, L., Eerola, K., Shevchouk, O. T., Richard, J. E., Nilsson, F. H., Jansson, L. E., Hayes, M. R., & Skibicka, K. P. (2020). The supramammillary nucleus controls anxiety-like behavior; key role of GLP-1R. Psychoneuroendocrinology, 119, 104720. https://doi.org/10.1016/j.psyneuen.2020.104720

Maniscalco, J. W., & Rinaman, L. (2017). Interoceptive modulation of neuroendocrine, emotional, and hypophagic responses to stress. Physiology & Behavior, 176, 195–206. https://doi.org/10.1016/j.physbeh.2017.01.027

Maniscalco, J. W., Zheng, H., Gordon, P. J., & Rinaman, L. (2015). Negative Energy Balance Blocks Neural and Behavioral Responses to Acute Stress by “Silencing” Central Glucagon-Like Peptide 1 Signaling in Rats. The Journal of Neuroscience: The Official Journal of the Society for Neuroscience, 35(30), 10701–10714. https://doi.org/10.1523/JNEUROSCI.3464-14.2015

Maske, C. B., Coiduras, I. I., Ondriezek, Z. E., Terrill, S. J., & Williams, D. L. (2020). Intermittent High-Fat Diet Intake Reduces Sensitivity to Intragastric Nutrient Infusion and Exogenous Amylin in Female Rats. Obesity (Silver Spring, Md.), 28(5), 942–952. https://doi.org/10.1002/oby.22779

Merchenthaler, I., Lane, M., & Shughrue, P. (1999). Distribution of pre-pro-glucagon and glucagon-like peptide-1 receptor messenger RNAs in the rat central nervous system. J Comp Neurol, 403, 261–280.

Mukherjee, A., Hum, A., Gustafson, T. J., & Mietlicki-Baase, E. G. (2020). Binge-like palatable food intake in rats reduces preproglucagon in the nucleus tractus solitarius. Physiology & Behavior, 219, 112830. https://doi.org/10.1016/j.physbeh.2020.112830

Mul, J. D., Begg, D. P., Barrera, J. G., Li, B., Matter, E. K., D’Alessio, D. A., Woods, S. C., Seeley, R. J., & Sandoval, D. A. (2013). High-fat diet changes the temporal profile of GLP-1 receptor-mediated hypophagia in rats. Am J Physiol Regul Integr Comp Physiol, 305(1), R68-77. https://doi.org/10.1152/ajpregu.00588.2012

Peng, J., Long, B., Yuan, J., Peng, X., Ni, H., Li, X., Gong, H., Luo, Q., & Li, A. (2017). A Quantitative Analysis of the Distribution of CRH Neurons in Whole Mouse Brain. Frontiers in Neuroanatomy, 11. https://doi.org/10.3389/fnana.2017.00063

Rinaman, L. (1999a). Interoceptive stress activates glucagon-like peptide-1 neurons that project to the hypothalamus. Am J Physiol, 277(2 Pt 2), R582-90.

Rinaman, L. (1999b). A functional role for central glucagon-like peptide-1 receptors in lithium chloride-induced anorexia. Am J Physiol, 277(5 Pt 2), R1537-40.

Terrill, S. J., Holt, M. K., Maske, C. B., Abrams, N., Reimann, F., Trapp, S., & Williams, D. L. (2019). Endogenous GLP-1 in lateral septum promotes satiety and suppresses motivation for food in mice. Physiology & Behavior, 206, 191–199. https://doi.org/10.1016/j.physbeh.2019.04.008

Terrill, S. J., Maske, C. B., & Williams, D. L. (2018). Endogenous GLP-1 in lateral septum contributes to stress-induced hypophagia. Physiology & Behavior, 192, 17–22. https://doi.org/10.1016/j.physbeh.2018.03.001

Vrang, N., Phifer, C. B., Corkern, M. M., & Berthoud, H. R. (2003). Gastric distension induces c-Fos in medullary GLP-1/2-containing neurons. American Journal of Physiology.Regulatory, Integrative and Comparative Physiology, 285, R470-8. https://doi.org/10.1152/ajpregu.00732.2002

Vrang, Niels, Hansen, M., Larsen, P. J., & Tang-Christensen, M. (2007). Characterization of brainstem preproglucagon projections to the paraventricular and dorsomedial hypothalamic nuclei. Brain Res, 1149, 118–126. https://doi.org/10.1016/j.brainres.2007.02.043

Williams, D. L., Hyvarinen, N., Lilly, N., Kay, K., Dossat, A., Parise, E., & Torregrossa, A. M. (2011). Maintenance on a high-fat diet impairs the anorexic response to glucagon-like-peptide-1 receptor activation. Physiol Behav, 103(5), 557–564. https://doi.org/10.1016/j.physbeh.2011.04.005

Williams, Diana L., Lilly, N. A., Edwards, I. J., Yao, P., Richards, J. E., & Trapp, S. (2018). GLP-1 action in the mouse bed nucleus of the stria terminalis. Neuropharmacology, 131, 83–95. https://doi.org/10.1016/j.neuropharm.2017.12.007

Yamamoto, H., Kishi, T., Lee, C. E., Choi, B. J., Fang, H., Hollenberg, A. N., Drucker, D. J., & Elmquist, J. K. (2003). Glucagon-like peptide-1-responsive catecholamine neurons in the area postrema link peripheral glucagon-like peptide-1 with central autonomic control sites. J Neurosci, 23(7), 2939–2946.

Yamamoto, H., Lee, C. E., Marcus, J. N., Williams, T. D., Overton, J. M., Lopez, M. E., Hollenberg, A. N., Baggio, L., Saper, C. B., Drucker, D. J., & Elmquist, J. K. (2002). Glucagon-like peptide-1 receptor stimulation increases blood pressure and heart rate and activates autonomic regulatory neurons. J Clin Invest, 110(1), 43–52. https://doi.org/10.1172/jci15595

Zhang, R., Jankord, R., Flak, J. N., Solomon, M. B., D’Alessio, D. A., & Herman, J. P. (2010). Role of glucocorticoids in tuning hindbrain stress integration. The Journal of Neuroscience: The Official Journal of the Society for Neuroscience, 30(44), 14907–14914. https://doi.org/10.1523/JNEUROSCI.0522-10.2010

Zheng, H., Reiner, D. J., Hayes, M. R., & Rinaman, L. (2019). Chronic Suppression of Glucagon-Like Peptide-1 Receptor (GLP1R) mRNA Translation in the Rat Bed Nucleus of the Stria Terminalis Reduces Anxiety-Like Behavior and Stress-Induced Hypophagia, But Prolongs Stress-Induced Elevation of Plasma Corticosterone. The Journal of Neuroscience: The Official Journal of the Society for Neuroscience, 39(14), 2649–2663. https://doi.org/10.1523/JNEUROSCI.2180-18.2019

Reviewer 2 Report

Summary:

The authors present a wide-ranging review of the literature describing the role of the gut hormone and neuromodulator glucagon-like peptide 1 in stress responses and feeding, particularly focused on integration of neural and endocrine signals. This an important subject, and readers within and outside the field would benefit from a comprehensive synthesis of the extensive literature relevant to this topic.  

Broad Comments:

The abstract contains a series of statements but rather lacks logical flow, and provides little conceptual overview or justification for why this review was written.

A number of statements are made regarding the GLP-1-producing PPG neurons in the brain, which are either confusing, inappropriately conflate peripheral and central GLP-1, and/or just do not accurately represent the literature on this topic. As an example, figure 1 implies that GLP-1R-expressing vagal afferent neurons innervate GLP-1R-expressing neurons in the AP, which then innervate GLP-1 producing neurons in the NTS to provide the sole neural input to the central GLP-1 system. The broad idea that gut-derived GLP-1 elicits central effects by activating PPG signalling is prevalent, however there is little (if any) evidence to support it, and emerging evidence suggests these are likely independent systems.  However, it is well established that the NTS, rather than the AP, is the primary target for vagal afferents, and direct vagal innervation of PPGs specifically has been demonstrated by electrophysiology (Hisadome et al 2010, Diabetes 59:1890) and retrograde tracing (Holt et al 2019, J Neuroscience 39:9767). In contrast, it is unclear what evidence the authors have in mind supporting the idea that GLP-1R neurons in the AP are innervated by vagal afferents, or provide input to PPG neurons. Overall, more care and precision is required with the scholarship on this topic.

While quite a detailed and comprehensive review of the literature, it is largely descriptive, with little in the way of a coherent narrative. In particular, for a review nominally describing ‘integration of neural and endocrine responses to feeding under stress’, neither integration of neural and hormonal signals nor the effects of stress on feeding seem particularly central to the manuscript as currently written. The manuscript would benefit from the sometimes disparate content being tied together into it’s central thesis more consistently.  

The standard of writing is quite variable throughout the manuscript. Sections 2.1 – 2.2.4 in particular are very clearly written, however other sections contain numerous spelling mistakes, inappropriately used technical wording, and grammatical errors, and so require comprehensive editing and proof reading.

Specific Comments:

P1 L15: NTS is the acronym for nucleus tractus solitarius – either use this or use the correct acronym.

P2 L54: Proglucagon is the product of the glucagon gene Gcg.

P2 L56-59: These two sentences describe the same population of neurons, which are typically described as preproglucagon (PPG) neurons, at least in mouse.

P2 L66: Should mention GLP-1RAs initially developed and licensed for T2D treatment.

P2 L68: ‘But also’ – effects on food reward are thought to be part of the mechanism of action.

Fig. 1: The colour scheme and arrows implies that vagal afferents are a type of endocrine signal – the label should be a blue box for consistency.

P3 L79: This is not really describing the physiological incretin effect.

P3 L85: Confusing logic – a substantial number of the ‘different effects’ of GLP-1 listed are mediated by neural pathways.

P3 L 102: Should cite Cork et al 2015, Molecular Metabolism 4:718.

P4 L142: Saying most actions are based in the hypothalamus is an overly strong statement.

P7 L259: ‘…Ex4 improves the levels of glutamatergic receptors (GluN1)…’ is imprecise – the paper cited describes Ex4-mediated reversal of diabetes-induced decrease in the GluN1 subunit content of NMDA receptors.

P7 L300: Unclear how this relates to figure 1.

P8 L332: The statement that the AP is the link for peripheral GLP-1 to activate central autonomic sites is an oversimplification – it is certainly a link, however vagal signalling to the NTS (or hormonal signalling via other circumventricular organs) cannot be ignored.

P9 L387: This paragraph appears to assume that central GLP-1 signalling is linked to gut GLP-1 actions, without any supporting evidence. It also conflates satiety and satiation.

Fig. 3: The logic of this figure is confusing (as is the use of red and green arrows when up and down ones would be more intuitive), and the information might be better summarised in a table, in which the link between sites of action and outcomes would be more clear.

P11 L437: What does ‘causing motivation / cognition’ mean?

P13 L132: Where are the citations for these statements? Also, the hypophagic effect of acute restraint stress in mice has been shown to be completely abolished by chemogenetic inhibition of PPG neurons – this is reference #13, and yet this very relevant finding has not been described in the manuscript.

P13 L575: What is meant by ‘almost clear’?

P14 L611: More accurate to say it does act as a neuromodulator.

P14 L615: Saying that GLP-1 ‘controls the stress response’ is an overly strong  and simplified claim.

P14 L620: The logic of this paragraph from this point on is hard to follow.

Formatting of the bibliography needs to be checked and standardised.